# Computed tomography shows high fracture prevalence among physically active forager-horticulturalists with high fertility

Jonathan Stieglitz[1,2]*, Benjamin C Trumble[3,4], HORUS Study Team, Caleb E Finch[5], Dong Li[6], Matthew J Budoff[7], Hillard Kaplan[8], Michael D Gurven[9]

[1]Université Toulouse 1 Capitole, Toulouse, France; [2]Institute for Advanced Study in Toulouse, Toulouse, France; [3]Center for Evolution and Medicine, Arizona State University, Tempe, United States; [4]School of Human Evolution and Social Change, Arizona State University, Tempe, United States; [5]Leonard Davis School of Gerontology, University of Southern California, Los Angeles, United States; [6]School of Medicine, Emory University, Atlanta, United States; [7]Los Angeles Biomedical Research Institute, Harbor-University of California at Los Angeles Medical Center, Torrance, United States; [8]Economic Science Institute, Chapman University, Orange, United States; [9]Department of Anthropology, University of California, Santa Barbara, Santa Barbara, United States

**Abstract** Modern humans have more fragile skeletons than other hominins, which may result from physical inactivity. Here, we test whether reproductive effort also compromises bone strength, by measuring using computed tomography thoracic vertebral bone mineral density (BMD) and fracture prevalence among physically active Tsimane forager-horticulturalists. Earlier onset of reproduction and shorter interbirth intervals are associated with reduced BMD for women. Tsimane BMD is lower versus Americans, but only for women, contrary to simple predictions relying on inactivity to explain skeletal fragility. Minimal BMD differences exist between Tsimane and American men, suggesting that systemic factors other than fertility (e.g. diet) do not easily explain Tsimane women's lower BMD. Tsimane fracture prevalence is also higher versus Americans. Lower BMD increases Tsimane fracture risk, but only for women, suggesting a role of weak bone in women's fracture etiology. Our results highlight the role of sex-specific mechanisms underlying skeletal fragility that operate long before menopause.
DOI: https://doi.org/10.7554/eLife.48607.001

*For correspondence:
jonathan.stieglitz@iast.fr

**Competing interests:** The authors declare that no competing interests exist.

## Introduction

Compared to other hominoids and extinct hominins, modern humans have postcranial skeletons that are more gracile (i.e. lower bone mass and strength for body size) (*Cotter et al., 2011*; *Ruff et al., 2015*; *Ryan and Shaw, 2015*). Declining skeletal strength has been documented in the cortical structure of long bone diaphyses (e.g. size or shape in a cross-section) and in trabecular bone micro-structure (e.g. thickness, bone volume fraction) and is particularly evident in the later Pleistocene or Holocene (*Chirchir, 2019*; *Chirchir et al., 2015*; *Ruff et al., 2015*; *Ryan and Shaw, 2015*). Physical inactivity is the most common explanation for skeletal fragility, based largely on the well-established principle that impact forces from load bearing and muscle contraction trigger bone deposition (*Carter and Orr, 1992*). According to this principle, which has been documented in numerous studies of competitive athletes and exercise interventions (e.g. *Polidoulis et al., 2012*; *Warden et al.,*

*2014*), and inferred from skeletal remains of prehistoric populations (e.g. *Macintosh et al., 2017*; *Ruff et al., 2015*), bone responds to physical activity demands by adding tissue and altering cross-sectional distribution in the direction of highest bending strains (i.e. change in length per unit length) (but see *Demes et al., 2001*; *Lieberman et al., 2004*; *Lovejoy et al., 2003* and references therein). This mechanical response of bone to loading is variable throughout the body, depending on types of weight-bearing activity and muscle function.

Evolutionary life history theory provides a broad explanatory framework that incorporates ultimate and proximate levels of analysis for understanding variability in bone strength (i.e. ability to withstand an applied load). In all organisms, limited resources are allocated to competing metabolic demands so as to optimize biological fitness. Due to higher fitness gains of reproduction earlier versus later in life (*Williams, 1957*), natural selection often prioritizes investments in earlier reproduction over somatic maintenance. Organisms may thus increase fertility at the expense of maintenance (*Kirkwood and Austad, 2000*), and we should expect greater energetic investments in reproduction to trade-off against investments in maintenance (*Stearns, 1992*). Consistent with life history theory, reproductive effort is expected to moderate effects of physical activity on adult bone strength, which is an indicator of energetic investment in somatic maintenance.

Tests of this life history trade-off in humans are inconclusive (*Le Bourg, 2007*), in part because many studies focus on mortality rather than investments in maintenance per se (but see *Ryan et al., 2018*; *Ziomkiewicz et al., 2016*), precluding direct analysis of whether greater reproductive effort inhibits maintenance. Bone tissue is ideal for examining metabolic trade-offs between reproduction and maintenance. Constant remodeling is necessary to maintain bone strength, and the skeleton is a general mineral reservoir for the competing metabolic demands of maternal maintenance and fetal bone accretion or lactation (*Stieglitz et al., 2015*). The average full-term human fetus has ~30 g calcium, 20 g phosphorus and 0.8 g magnesium, and at least 80% of these macro-minerals are accreted in the third trimester (see *Kovacs, 2016* for a comprehensive review of bone metabolism during pregnancy, lactation and post-weaning). For an average-sized fetus this corresponds at week 24 of gestation to a mean calcium [phosphorus] transfer rate of ~60 mg/day [~40 mg/day] and between weeks 35–40 of 300–350 mg/day [200 mg/day]. In the third trimester, hourly fetal transfers of calcium and phosphorus are between 5 and 10% of that present in maternal plasma, which is enough to provoke maternal hypocalcemia and hypophosphatemia. Generally, patterns of bone turnover are similar comparing pre- to early pregnancy states, but turnover increases during the third trimester to create a net resorptive state. During the first six months [second six months] postpartum, ~200 mg/day [~120 mg/day] of calcium is secreted into human breast milk. Analyses of bone turnover markers, bone mineral density (BMD), and bone structure by high-resolution peripheral quantitative computed tomography (HR-pQCT) suggest that lactating women are in negative calcium balance, especially when milk production is elevated. Longitudinal studies show consistent declines in lactating women's BMD or bone mineral content, with mean declines of 3–10% after 3–6 months of lactation. The greatest BMD losses (5–10%) occur in the lumbar spine, with modest losses (<5%) occurring at sites with less trabecular bone, and smaller losses (<2%) at sites containing mostly cortical bone. The few studies utilizing HR-pQCT in the limbs (radius, femur) show smaller (<2%) reductions in trabecular thickness and cortical thickness and volume, with greater reductions among women who lactate longer. Thus, because mineral allocations to maternal maintenance and reproduction draw from the same skeletal reservoir, direct trade-offs between these competing demands should manifest in bone. These trade-offs are expected to manifest in the longer term, regardless of whether maternal bone tissue fully or only partially recovers from mineral losses following a specific bout of gestation and lactation. Life history theory makes no assumptions or predictions about the extent of bone mineral recovery (i.e. whether full or partial) after weaning a specific child. Accordingly, a general hypothesis from life history theory is that greater reproductive effort constrains the ability of bone tissue to respond to mechanical loading and high physical activity levels (PALs). This hypothesis is not an alternative to and may complement other hypotheses of bone structural variation emphasizing developmental factors affecting the trade-off between investment in growth and reproduction (*Macintosh et al., 2018*). But unlike other hypotheses of bone structural variation derived from life history theory or proximate explanations (focusing, for example on nutrition, inflammation, hormones), the hypothesis emphasizing effects of reproductive effort uniquely predicts sex differences within and between populations, the magnitude of which should be influenced by relative investment in reproduction.

Timing of reproduction, in addition to lifetime reproductive effort, is also expected to affect bone strength (*Macintosh et al., 2018*). Peak bone mass is typically not achieved until the late 20s, so earlier pregnancy and lactation can disrupt maternal bone growth and/or mineralization (*Madimenos et al., 2012*; *Stieglitz et al., 2015*), potentially reducing peak bone mass and thus later-life bone strength (but see *Chantry et al., 2004*). In addition to early onset of reproduction, short interbirth intervals (IBIs) can potentially generate unbalanced cycles of maternal bone resorption and formation, limiting maternal skeletal recovery before subsequent pregnancy (*Stieglitz et al., 2015*). Whether the maternal skeleton is fully or partially restored post-weaning to its prior mineral content and strength, despite potentially lasting micro-architectural changes shown in recent imaging studies, is currently debated (*Clarke and Khosla, 2010*; *Kovacs, 2016*; *Kovacs, 2017*; *Wysolmerski, 2010*) and physiological mechanisms underlying post-weaning maternal skeletal recovery are not well understood. Dual-energy X-ray absorptiometry data suggest that for most women, lactation-associated BMD losses and micro-architectural deficits are reversed by 12 months post-weaning, although patterns can vary by skeletal site and most studies are conducted among well-nourished women with low fertility (*Kovacs, 2016*). Restorative capacity partly depends on lactation duration; HR-pQCT studies of the radius and femur show recovery of trabecular and cortical micro-architecture in women lactating for shorter periods, but incomplete recovery in women lactating for longer periods. A recent HR-pQCT study found incomplete recovery of trabecular and cortical micro-architecture in the tibia and radius after a median of 2.6 years post-weaning for women exclusively breastfeeding for five months (*Bjørnerem et al., 2017*). Incomplete restoration of lumbar spine BMD to pre-pregnancy values by 12 months post-partum is observed among rural Gambian women practicing on-demand breastfeeding for about two years (*Jarjou et al., 2010*).

A relevant literature on 'maternal depletion syndrome' (*Jelliffe and Maddocks, 1964*) examines trade-offs between reproductive effort and maternal health more broadly in high fertility contexts. Currently, evidence for maternal depletion is mixed (*Gurven et al., 2016*; *Tracer, 2002*), though most prior studies focus only on the period covering one or two births rather than the cumulative long-term effects on health of repeated pregnancies. Moreover, most of these prior studies focus on maternal anthropometric status (e.g. weight, adiposity) rather than bone tissue per se. Bone tissue fluctuates less than anthropometric markers with short-term changes in energy balance, rendering studies of bone less susceptible to sampling biases.

Here, we examine in a natural fertility population, Tsimane forager-horticulturalists of Bolivia, whether greater reproductive effort compromises bone strength, particularly for women given their greater energetic costs of reproduction. Tsimane are an ideal population to test whether women's greater reproductive effort compromises bone strength. Tsimane fertility is high (total fertility rate = 9 births per woman), birth spacing is short (*Stieglitz et al., 2015*), breastfeeding is on-demand, effective birth control is rare, and PALs are high (*Gurven et al., 2013*), as is typical of other small-scale rural subsistence populations. In a population-representative sample of adults aged 40+ years, who mostly have completed their reproduction, we utilize thoracic computed tomography (CT) to measure two primary indicators of bone strength in thoracic vertebrae: BMD, which accounts for ~70% of the variance in bone strength (*NIH Consensus Development Panel on Osteoporosis Prevention, Diagnosis, and Therapy, 2001*), and fracture prevalence and severity. We focus on thoracic vertebrae since spontaneous thoracic vertebral fractures are among the most common osteoporosis-related fractures in humans (*Sambrook and Cooper, 2006*). Such fractures have not been observed in wild or captive apes, even in individuals with severe osteopenia (*Gunji et al., 2003*), suggesting that modern humans are more susceptible than other primates to osteoporosis-related fractures (*Cotter et al., 2011*).

Most activities of daily living, including sitting, walking, running and lifting, generate loads on human vertebrae (*Myers and Wilson, 1997*; *Rohlmannt et al., 2001*; *Stewart and Hall, 2006*), and thus thoracic vertebrae track mechanics of both lower and upper limbs. Even regular breathing appears to generate intradiscal pressure and some loading in the thoracic spine (*Polga et al., 2004*). Vertebral bodies, which are composed mostly of trabecular bone surrounded by a thin cortical shell, function largely as shock absorbers and can deform to a greater degree than tubular bones (*Seeman, 2008*); deformation facilitates spinal flexion, extension and rotation. The major loading mode on human vertebral bodies is axially compressive, and most axial force is carried by the trabecular bone (*Myers and Wilson, 1997*). For many activities (e.g. neutral standing, standing with weight, mild trunk flexion and extension, lifting objects above the head), the greatest compressive loads

along the spine are generated in the thoracolumbar region (*Bruno et al., 2017*). The greatest compressive vertebral loads occur during activities in which body mass or externally applied weights are shifted anteriorly, such as during trunk flexion or carrying weight in front of the body (*Bruno et al., 2017*; *Polga et al., 2004*; *Rohlmannt et al., 2001*). To accommodate forces, the architecture of the vertebral body trabecular bone consists of thick vertical plates and columns supported by thinner horizontal trabeculae. This trabecular structure changes with age, such that vertical plates are successively perforated during remodeling and converted into columns, whereas horizontal trabeculae perforate and disappear (*Mosekilde et al., 1987*). These age-related trabecular structural changes can result in vertebral strength declines that are greater than predicted from bone mass estimation alone. Vertebral strength is compromised more by loss of trabecular connectivity than by trabecular thinning, and women are more susceptible than men to age-related horizontal trabecular perforation and disappearance (*Mosekilde, 1989*).

In this paper, we first test, among Tsimane women, whether greater reproductive effort – indicated by earlier age at first birth, higher parity and shorter IBIs – is associated with reduced thoracic vertebral BMD. We then test whether Tsimane BMD is lower, particularly for women, than a matched American sample with directly comparable CT-derived indicators of bone strength. This latter prediction follows from the hypothesis that greater reproductive effort compromises bone strength, which is consistent with a trade-off between investment in reproduction and maintenance as posited by evolutionary life history theory. In contrast, a simple prediction from a physical inactivity hypothesis for compromised bone strength posits the opposite, that is, lower BMD for Americans than Tsimane, for both sexes, given lower PALs, on average, among Americans. Regarding the second measured bone strength indicator, thoracic vertebral fracture, we test whether Tsimane fracture prevalence is higher than matched Americans, particularly for women. To determine whether Tsimane thoracic vertebral fracture results from compromised bone strength as opposed to trauma, we test whether fracture risk is inversely associated with thoracic vertebral BMD. Lastly, we test whether Tsimane women's fracture risk increases with reproductive effort, even after adjusting for BMD, which is expected if greater reproductive effort compromises bone micro-architecture in complex ways beyond just reducing mineral density (e.g. by reducing trabecular thickness or connectivity density).

## Results

### Tsimane women's thoracic vertebral BMD declines with early age at first birth and short IBI

Earlier age at first birth is associated with reduced BMD (Std. $\beta_{Age\ at\ 1st\ birth\ [years,\ logged]}$=0.099, p=0.036, controlling for age and fat-free mass, adj. $R^2$ = 0.51, n = 213; *Appendix 1—table 1*). Back-transforming logged age at first birth values into observed values and holding controls at sample means, there is a BMD difference of 0.57 SDs for women with maximum versus minimum age at first birth (37 versus 12 years, respectively).

Parity (continuously or categorically operationalized) is not associated with BMD controlling for age at first birth, age and fat-free mass (*Appendix 1—table 2*), nor does parity interact with any indicator of reproductive effort to predict BMD.

Shorter mean IBI (<29.7 months) is associated with lower BMD (Std. $\beta_{Shorter\ mean\ IBI}$=-0.201, p=0.032, controlling for age at first birth, age and fat-free mass, adj. $R^2$ = 0.52; *Appendix 1—tables 3–4*). Mean IBI also interacts with age at first birth: BMD is 0.28 SDs higher for women with longer mean IBI and later age at first birth versus women with shorter mean IBI and earlier age at first birth (interaction p=0.027, controlling for age and fat-free mass; see *Figure 1* and *Appendix 1—figure 1*).

Additionally controlling for indicators of modernization, that is, residential proximity to the closest market town of San Borja, Spanish fluency and schooling, which could reflect differential activity levels, diet and/or other factors affecting bone strength (e.g. infectious burden), strengthens the association between BMD and both mean IBI and age at first birth (comparing estimates in *Appendix 1—table 5* to those in *Appendix 1—table 4*). BMD is not significantly associated with any modernization indicator. Neither young age at menarche nor menopause is associated with BMD

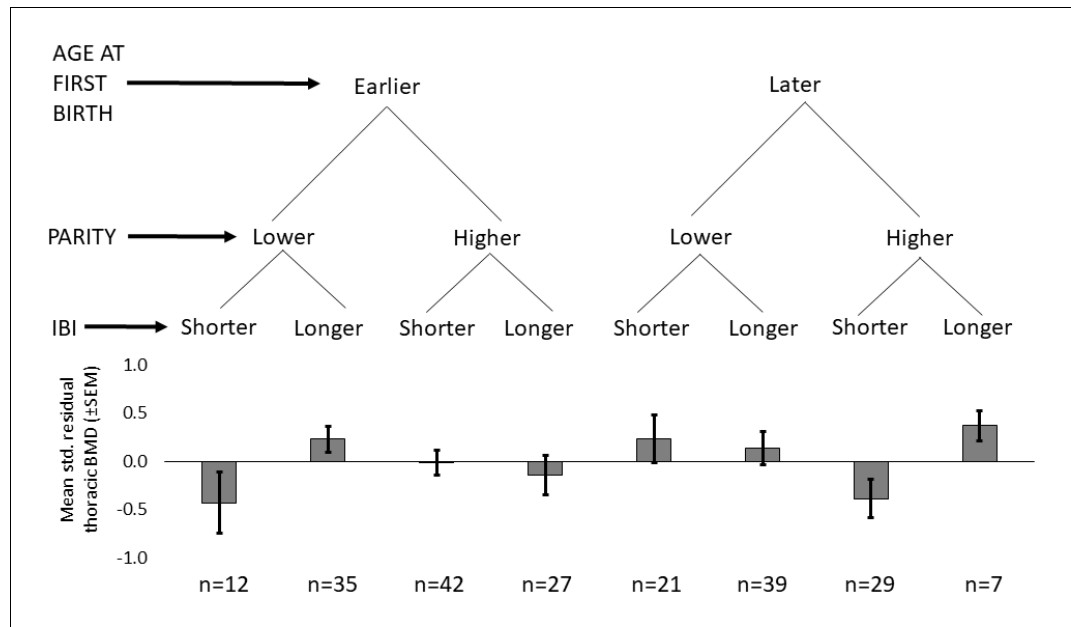

**Figure 1.** Standardized residual thoracic vertebral BMD (mean ±SEM; controlling for age and fat-free mass) by reproductive effort (n = 212 Tsimane women). Earlier vs. later age at first birth is defined as <18 vs. ≥18 years (median split), respectively; lower vs. higher parity is defined as ≤9 vs. >9 live births (median split), respectively; and shorter vs. longer mean IBI is defined as <29.7 months vs. ≥29.7 (median split), respectively.
DOI: https://doi.org/10.7554/eLife.48607.002

(*Appendix 1—table 6*), nor does either variable interact with any indicator of reproductive effort to predict BMD.

## Thoracic vertebral BMD is lower for Tsimane than Americans, but only for women

For women, age-standardized mean BMD is 8.9% lower for Tsimane than Americans (*Figure 2*; *Appendix 1—table 7*). BMD is significantly lower (all p's < 0.01) for Tsimane than Americans at all

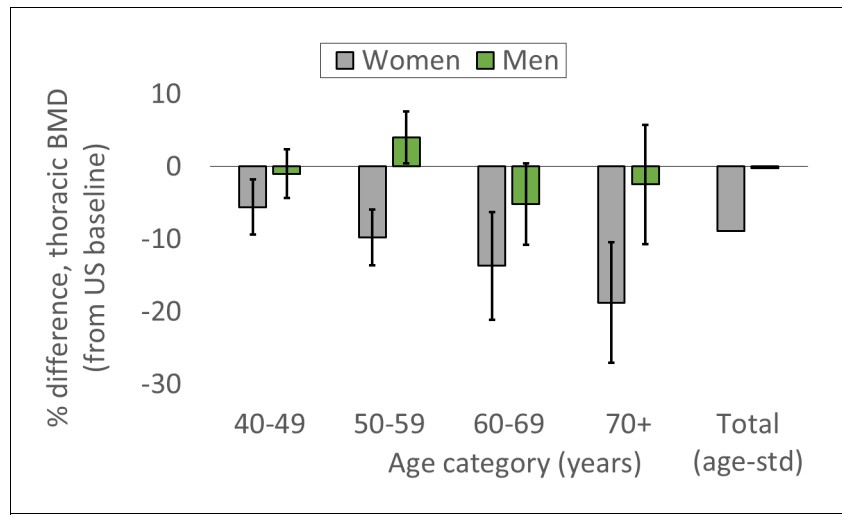

**Figure 2.** Tsimane versus American mean thoracic vertebral BMD (95% CIs) by age and sex. The 'total' category is age-standardized (see *Appendix 1—table 7* for values and details).
DOI: https://doi.org/10.7554/eLife.48607.003

ages, and this population-level difference increases with age in the cross-section. Mean BMD for Tsimane women is 5.6% lower than age-matched Americans aged 40–49 years, 9.8% lower for ages 50–59, 13.7% lower for ages 60–69, and 18.8% lower for women aged 70+. For men, age-standardized mean BMD is 0.24% lower for Tsimane than Americans. Tsimane BMD is not significantly different at ages 40–49 (1.0% lower for Tsimane, p=0.547), 60–69 (5.2% lower for Tsimane, p=0.069) and age 70+ (2.5% lower for Tsimane, p=0.543); Tsimane men aged 50–59 have 4% higher mean BMD (p=0.027) than age-matched Americans.

## Tsimane thoracic vertebral fracture prevalence is higher than Americans

For women, age-standardized prevalence of any thoracic vertebral fracture (i.e. grade ≥1; including mild, moderate or severe) for Tsimane and Americans is 18% and 9%, respectively (adjusted $RR_{Tsimane\ [vs.\ US]}$=1.81, 95% CI: 1.16–2.83, p=0.009, controlling for age, n = 491; see *Figure 3* and *Appendix 1—table 8*). Using a more conservative fracture definition (i.e. grade ≥2; including only moderate or severe), age-standardized prevalence for Tsimane and Americans is 6% and 2%, respectively (adjusted $RR_{Tsimane\ [vs.\ US]}$=2.69, 95% CI: 1.07–6.75, p=0.035; *Appendix 1—table 9*). For men, age-standardized prevalence of any fracture for Tsimane and Americans is 36% and 11%, respectively (adjusted $RR_{Tsimane\ [vs.\ US]}$=3.30, 95% CI: 2.26–4.82, p<0.001, n = 524; *Appendix 1—table 8*), and using a more conservative fracture definition, 10% and 2%, respectively (adjusted $RR_{Tsimane\ [vs.\ US]}$=3.79, 95% CI: 1.78–8.09, p=0.001; *Appendix 1—table 9*).

For both sexes, Tsimane are significantly more likely than Americans to present moderate (i.e. grade 2) but not severe (grade 3) fracture (*Appendix 1—tables 10–11*). Tsimane men but not women are significantly more likely than Americans to present borderline deformity (grade 0.5) and mild fracture (grade 1).

## Lower thoracic vertebral BMD increases risk of thoracic vertebral fracture among Tsimane, particularly for women. Short IBI additionally increases Tsimane women's fracture risk

Tsimane women with thoracic vertebral fracture (grade ≥1) have lower BMD (0.46 SDs; p=0.005), higher parity (0.25 SDs; p=0.071) and shorter mean IBI (0.22 SDs; p=0.011) than women without fracture (*Appendix 1—table 12*). There are no significant differences between women without versus with fracture in terms of age, anthropometrics, ages at menarche, menopause or first birth, or modernization indicators. Women's BMD is inversely associated with fracture risk (adjusted $RR_{BMD}$ per SD increase = 0.542, 95% CI: 0.352–0.837, p=0.006, controlling for age, height and fat mass, n = 219); this association remains (adjusted $RR_{BMD}$ = 0.540, 95% CI: 0.345–0.845, p=0.007; *Figure 4*)

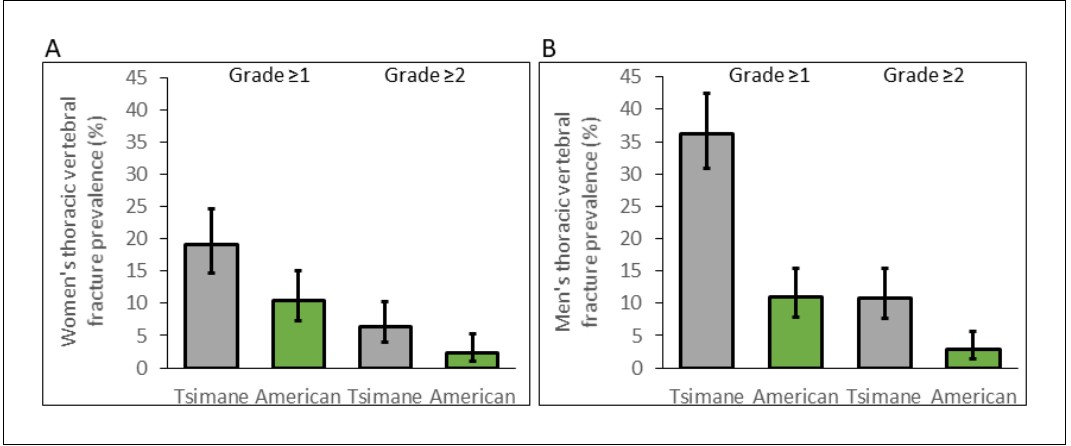

**Figure 3.** Thoracic vertebral (T6–T12) fracture prevalence (95% CIs) for Tsimane (in gray) and American (in green) women (**A**) and men (**B**) by fracture grade. Prevalence based on a less (grade ≥1) and more (grade ≥2) conservative fracture definition is shown. Prevalence is estimated from log-binomial generalized linear models adjusting for age.
DOI: https://doi.org/10.7554/eLife.48607.004

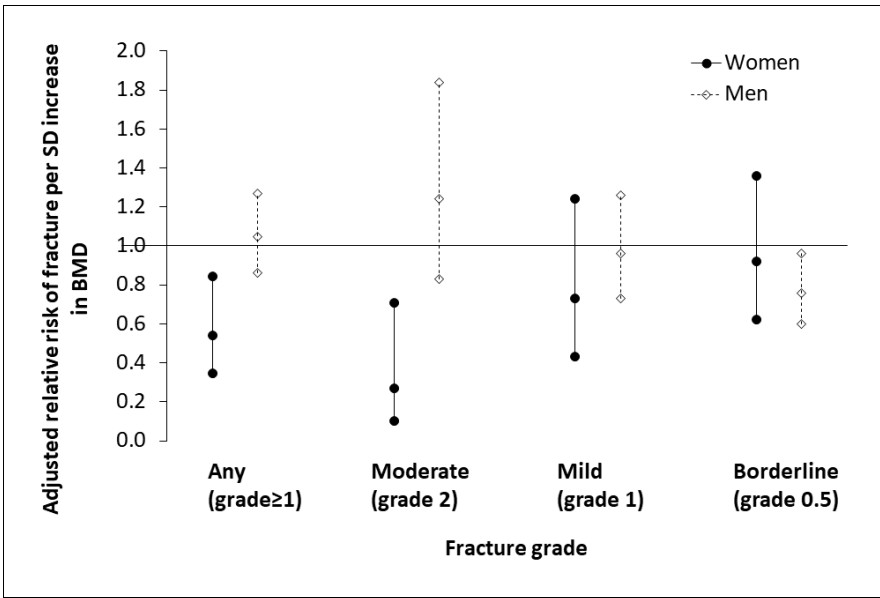

**Figure 4.** Association between thoracic vertebral BMD and thoracic vertebral (T6–T12) fracture risk (95% CI) for Tsimane. Log-binomial generalized linear models are used to estimate relative risk after adjustment for age, height and fat mass. Mean IBI is also included as a covariate for women. Parameter estimates are shown in *Appendix 1—tables 13–14* (women), and *Appendix 1—tables 16–17* (men). Severe fractures (grade 3) are omitted due to their relative scarcity.
DOI: https://doi.org/10.7554/eLife.48607.005

after adding to the model mean IBI, which is also inversely associated with fracture risk (adjusted $RR_{Mean\ IBI}$ per SD increase = 0.379, 95% CI: 0.165–0.866, p=0.021; *Appendix 1—table 13*: Model 2). The inverse association between women's BMD and fracture risk strengthens with higher fracture grades (*Appendix 1—table 14*), whereas the inverse association between mean IBI and fracture risk is strongest for mild (i.e. grade 1) fracture. In multivariate models, we found no effect on women's fracture risk of either parity, ages at menarche, menopause or first birth, or modernization indicators (controlling for BMD, mean IBI, age, height and fat mass).

Tsimane men with fracture are shorter (0.23 SDs; p=0.038), have higher adiposity (0.37 SDs; p=0.025) and have higher BMI (0.34 SDs; p=0.012) than men without fracture (*Appendix 1—table 15*). As with women, men without versus with fracture do not significantly differ in terms of age, weight, fat-free mass or modernization indicators. But unlike women, there is no significant difference in BMD between men without versus with fracture, and men's BMD is not associated with risk of any fracture (adjusted $RR_{BMD}$ per SD increase = 1.046, 95% CI: 0.862–1.269, p=0.650, controlling for age, height and fat mass, n = 227; *Appendix 1—table 16*). Men's BMD is, however, inversely associated with risk of borderline (i.e. grade 0.5) deformity (*Appendix 1—table 17*; *Figure 4*). Modernization indicators are not associated with men's fracture risk, nor do they meaningfully moderate associations between BMD and fracture risk.

## Discussion

This paper makes five empirical contributions. We find that: 1) Tsimane thoracic vertebral BMD is lower among women with early age at first birth and short IBIs; 2) Tsimane BMD is lower than a directly comparable American sample, but only for women; 3) thoracic vertebral fracture prevalence is higher for Tsimane than Americans; 4) among Tsimane, lower BMD is associated with higher fracture risk, particularly for women; and 5) short IBIs are associated with Tsimane women's fracture risk even after adjusting for BMD.

These results are consistent with a life history trade-off between reproductive effort and somatic maintenance (*Kirkwood and Austad, 2000*; *Stearns, 1992*; *Stieglitz et al., 2015*). Maternal physiology uniquely responds to the greater mineral demands of pregnancy and lactation by mobilizing

skeletal mineral stores (*Kovacs, 2016*; *Nelson et al., 2002*; *Prentice, 2003*). Maternal regulatory mechanisms can compensate for acute mineral losses, for example, by retaining excess mineral in circulation to facilitate storage. But in energy-limited settings (due to high energy expenditure relative to consumption), earlier age at first birth and shorter birth spacing can potentially reduce peak bone mass and generate unbalanced cycles of bone resorption and formation, compromising bone strength long before menopause. In addition to direct negative effects of reproductive effort on bone strength, indirect negative effects are possible via reductions in women's PALs and mechanical loading on bone, given nursing women's reduced food acquisition efforts relative to non-nursing women (cf. *Hurtado et al., 1992*). Tsimane women's participation in certain subsistence activities ostensibly entailing high-impact and high-magnitude loadings (e.g. rice-pounding using thick wooden bats) may be curtailed during pregnancy and early lactation; meta analyses of controlled exercise trials indicate that high-impact and high-magnitude loadings are especially effective at preserving women's BMD at the lumbar spine, femoral neck and total hip (*Martyn-St James and Carroll, 2009*; *Martyn-St James and Carroll, 2010*). In the present study, we also found that parity per se is not inversely associated with BMD (*Appendix 1—tables 2–3*; but see *Stieglitz et al., 2015*). Prior studies of parity-specific effects on maternal BMD do not reveal consistent associations (*Kovacs, 2016*), although most studies are conducted in developed countries with lower fertility and greater energetic surpluses. These conditions can obscure expected energetic trade-offs between reproduction and maintenance.

The finding that thoracic vertebral BMD is lower for Tsimane than in a comparable sample of American women, whereas for men few population-level differences are apparent, despite higher mean PALs of Tsimane women and men, contributes to a growing literature emphasizing sexual dimorphism in skeletal responses to environmental stimuli (e.g. *Macintosh et al., 2017*), mediated in part by sex and growth hormones, growth factors and their receptors (*Devlin, 2011*; *Gosman et al., 2011*). Dietary or other systemic population-level differences (e.g. in infectious burden) do not easily explain Tsimane women's lower BMD because such systemic factors should affect both sexes. Recent dietary analyses of Tsimane and Americans (NHANES) based on 24 hr dietary recalls indicate minor sex differences in proportional macronutrient intake for both populations (*Kraft et al., 2018*). In absolute terms, Tsimane mean daily per-capita energy, protein and carbohydrate intake actually exceeds that of Americans for both sexes, as does Tsimane intake of the bone-forming minerals magnesium, phosphorus and zinc. Nevertheless, prior Tsimane research indicates that energetic limitation and greater immune activation from high pathogen exposure partly explain why mean PALs above those of industrialized societies do not inevitably yield elevated peak bone mass or protect against age-related bone loss for either sex (*Stieglitz et al., 2015*; *Stieglitz et al., 2016*). In a prior cross-sectional study of women utilizing ultrasound, we found reduced calcaneal strength for Tsimane versus Americans that is already apparent in the 20s, with population-level differences increasing with age (*Stieglitz et al., 2015*). Between- and within-population analyses in the present study suggest that sex-specific costs of reproduction contribute to skeletal sexual dimorphism, and that sexually dimorphic responses to reproduction manifest early in adulthood, depending on age at first birth and birth spacing (*Madimenos et al., 2012*; *Stieglitz et al., 2015*).

That Tsimane fracture prevalence is higher than a comparative American sample (*Figure 3* and *Appendix 1—tables 8–11*) may be surprising in light of the presumably protective higher lifetime moderate physical activity of Tsimane (*Gurven et al., 2013*), and their minimal exposure to other osteoporosis risk factors found in industrialized societies (e.g. glucocorticoid therapy, excessive smoking or alcohol consumption). By restricting the Tsimane sample to older adults (aged 40+ years), we minimize potential for cohort effects related to changing lifestyles associated with modernization. This is supported by the fact that study findings hold even after controlling for modernization indicators, which are not associated with thoracic vertebral BMD or fracture risk (*Appendix 1—tables 5–6*, *12*, *15*). Higher fracture prevalence among Tsimane versus Americans is noteworthy in light of relatively scant bio-archaeological evidence of osteoporotic fracture prior to industrialization (*Agarwal, 2008*; *Curate et al., 2010*) and evidence of increasing age-specific osteoporotic fracture incidence rates over time in Western populations (*Cooper et al., 2011*). But these results are not surprising in light of high Tsimane fertility and life history trade-offs between energetic investments in reproduction and bone growth and/or maintenance, and other factors (e.g. calcium deficiency, chronic immune activation due to high pathogen exposure), which may interact with

high reproductive effort to further constrain the ability of bone tissue to respond to mechanical loading and high PALs (cf. *Armelagos et al., 1972*).

The fact that Tsimane women's vertebral BMD is inversely associated with vertebral fracture risk (cf. *Mays, 1996*) (*Figure 4*) suggests a major role of compromised bone strength in precipitating fracture, rather than traumatic injury of otherwise healthy bone. For Tsimane men, it is likely that trauma plays a major role in precipitating fracture due to: 1) a weak association between BMD and fracture risk (*Figure 4*); 2) a high fracture prevalence relative to a comparative American sample (*Figure 3*) despite minimal BMD differences (*Figure 2*); and our anecdotal observations of high levels of mechanical stress on Tsimane men's vertebrae from frequent heavy load carrying (e.g. of hunted game, timber for constructing houses). This of course does not preclude a contributing role of compromised bone strength (e.g. from micro-architectural deficiencies) in precipitating Tsimane men's fracture. Likewise, for women, the present results do not preclude a contributing role of trauma in precipitating fracture. Future research is needed to determine the extent to which women's subsistence involvement and frequent carrying of young children and other loads (e.g. woven bags filled with harvested cultigens) influence bone structural integrity. Some of the greatest compressive vertebral loads occur when weights are carried in front of the body (*Bruno et al., 2017*; *Rohlmannt et al., 2001*), which is how Tsimane women routinely carry infants and toddlers. The fact that Tsimane women's shorter IBIs predict increased fracture risk even after adjusting for BMD, which also remains a significant predictor (*Appendix 1—table 13–14*), suggests that shorter IBIs compromise multiple aspects of vertebral micro-architecture (e.g. trabecular thickness or connectivity density), although further research is needed to examine this possibility.

## Inferring behaviors underlying morphological variation in past human populations

While our results do not directly address debates over the timing of and prior selection pressures underlying transition to skeletal gracility in past human populations, our results do provide insight into lifestyle factors affecting bone strength which may have been relevant during this transition. It has been hypothesized that subsistence transition from hunting and gathering to more sedentary agriculture, and increasing reliance on labor-saving technology, caused reductions in mechanical loading on bone and PALs, leading to modern human skeletal gracilization (*Chirchir, 2019*; *Chirchir et al., 2015*; *Chirchir et al., 2017*; *Ruff et al., 2015*; *Ryan and Shaw, 2015*). Increasing reliance on agriculture may have been associated with reductions in terrestrial mobility, and thus reduced mechanical loading of lower limbs, although the pace and magnitude of these changes likely varied temporally and spatially. Changes in upper body activities may have been much more variable during subsistence transitions, so upper limb loading may have actually increased with agriculture in some regions (*Bridges, 1989*). Nevertheless, comparisons of skeletal remains of hunter-gatherers and either full- or part-time agriculturalists indicate among agriculturalists reduced femoral strength, as indicated by trabecular bone structure or external size dimensions (*Larsen, 1981*; *Ryan and Shaw, 2015*), and accelerated age-related decline in radial bone mineral content (*Perzigian, 1973*).

Yet a physical inactivity explanation for modern human skeletal gracility – rooted in a subsistence transition from foraging to farming – is puzzling for several reasons. Evidence that agriculturalists are more sedentary than hunter-gatherers is not particularly strong: PALs and time allocation to work vary substantially within a subsistence regime, and both measures are actually higher among agriculturalists (*Gurven et al., 2013*; *Leonard, 2008*). Moreover, children in agricultural societies generally begin work earlier than hunter-gatherer children (*Kramer, 2005*). This is significant because higher PALs in childhood and early adulthood, particularly for higher-impact activities producing high peak stresses, increase peak bone mass, size and later-life bone strength (e.g. see *Warden et al., 2014* and references therein). Furthermore, in a prospective Tsimane study we found that time spent in horticulture positively predicted ultrasound-derived indicators of radial strength (*Stieglitz et al., 2017*), whereas tibial strength was not predicted by extent of involvement in any subsistence activity including hunting. While various studies show reduced lower limb strength among agriculturalists relative to pre-agriculturalists, as indicated by cross-sectional diaphyseal structure of cortical bone or trabecular bone volume fraction or thickness (*May and Ruff, 2016*; *Saers et al., 2016*), other studies using similar diaphyseal structural properties or external size dimensions show no differences in limb strength or dissimilar patterns by subsistence regime (*Bridges, 1989*; *Ruff, 1999*). Taken together,

these observations create uncertainty over the timing of and selection pressures underlying modern human skeletal gracility. This uncertainty is exacerbated given evidence of sex-specific skeletal responses to mechanical loading and physical activity (*Bridges, 1989*; *Macintosh et al., 2017*; *Ruff, 1999*). Additional sex-specific factors may have contributed to skeletal gracility during transition to farming, either independently and/or in interaction with PALs.

Our results instead suggest that fertility increases associated with subsistence transition from foraging to farming (*Bentley et al., 1993*) contributed to modern human skeletal gracility, particularly for women. Numerous proximate determinants have been proposed to explain fertility increases that accompanied greater energetic surpluses from agriculture, including earlier age of menarche and first birth, and shorter IBIs (e.g. due to earlier weaning and supplementary infant feeding) (e.g. *Campbell and Wood, 1988*). Regardless of proximate fertility determinants, given the trade-off between energetic investment in reproduction and somatic maintenance, our results suggest that women's skeletons were especially susceptible to gracilization during subsistence transition from foraging to farming in light of the associated increases in fertility, and reduced mobility resulting from increased fertility (cf. *Hurtado et al., 1992*). Sex-specific mechanisms beyond menopause underlying skeletal gracility during this subsistence transition should thus be considered, in addition to explanations emphasizing increased sedentism, reliance on labor-saving technology and associated reductions in mechanical loading on bone. Nevertheless, since skeletal gracilization is observed in both sexes during this subsistence transition (*Ruff et al., 2015*), any explanation solely resting on changes in reproductive effort cannot explain these morphological changes in men.

## Study limitations

The cross-sectional study design using retrospective demographic data limits our ability to establish that greater reproductive effort causes BMD reductions and fracture. The results presented here may thus be consistent with alternative interpretations derived from life history theory or proximate explanations, although such alternatives must address the observed population-level sex differences. Another limitation is that our measures of reproductive effort are indirect measures of reproductive costs, and we lack data on lactation duration or intensity. However, all Tsimane women breastfeed their infants and it is common for women to breastfeed exclusively for about four months (no study participant bottle fed an infant). We also lack estimates of vertebral size and geometry, which affect vertebral strength, and we lack individual-level data on activity level, nutrient intake and pathogen burden for population-level comparisons of BMD and fracture prevalence and severity. We also do not consider whether genetic diversity determines heterogeneity in BMD or fracture prevalence. Bone strength indicators such as BMD and fracture are heritable and the frequency of alleles affecting BMD differs between ethnically distinct populations (see *Wallace et al., 2016* and references therein). But while genetic factors can account for a sizable portion of variance in bone strength within populations, there is little evidence that heterogeneity in bone strength between populations is due to stochastic genetic diversity.

## Conclusion

This study examines direct indicators of bone strength from a clinically and mechanically relevant anatomic region using in vivo imaging in a physically active population with high fertility. Results suggest a trade-off between reproductive effort and bone strength, and that greater reproductive effort constrains the ability of bone tissue to respond to mechanical loading and high physical activity. Results also raise the possibility that increased fertility associated with subsistence transition from foraging to farming promoted modern human skeletal gracility, particularly among women. Because of the complex nature of lifestyle transformations during subsistence transitions, including apparent increases in infectious disease, nutritional deficiencies and dental decay (*Cohen and Armelagos, 1984*), an expanded conceptual framework incorporating diverse lifestyle factors that may constrain the ability of bone to respond to mechanical loading (e.g. high fertility, nutrient deficiency, infection-induced inflammation (*Madimenos et al., 2012*; *Stieglitz et al., 2015*; *Stieglitz et al., 2016*) can improve our understanding of morphological transformations associated with transition to farming. Of course, our ability to make inferences about the past using data collected in contemporary populations is limited. Tsimane are neither 'pure' hunter-gatherers nor agriculturalists, and they may differ in important ways from ancestral human populations in terms of residential mobility, fertility, diet

and disease exposures. Yet no single population represents the range of experiences across different environments that shaped the evolution of our species over the millennia in which ecologies fluctuated. In vivo study of bone strength in well-characterized, population-representative, non-industrialized societies provides an opportunity to examine lifestyle factors that are often invisible to bio-archaeological inquiry but nonetheless relevant to understanding selection pressures over human history.

## Materials and methods

### Study population

Tsimane forager-horticulturalists of lowland Bolivia are semi-sedentary and live in >90 villages. Their diet consists of cultigens grown in small swiddens (62% of total calories; mostly plantains [60% of cultigen-derived calories], rice, sweet manioc and corn), freshwater fish (16%), meat from hunting and domesticated animals (14%), market foods (8%; mostly pasta [55% of market-derived calories]) and wild fruit and vegetables (<1%) (*Kraft et al., 2018*). Relative to Western dietary recommendations calcium intake is low (~240 mg/day), but intake of other bone-forming minerals is ample (magnesium: ~525 mg/day; zinc: ~14 mg/day) or high (phosphorus: ~1,550 mg/day). Women's PAL is in the 'moderate to active' range (PAL = 1.7–1.9) and remains constant throughout adulthood. Men's PAL is 'vigorously active' (PAL = 2.0–2.2), and declines by 10–20% from the peak (in the late 20s) to older adulthood (age 60+ years) (*Gurven et al., 2013*).

### Study design and participants

The Tsimane sample includes all individuals who met the inclusion criteria of self-identifying as Tsimane and who were aged 40+ years (n = 507; 48% female; age range: 41–94 years; see Appendix for additional details and *Appendix 1—table 18* for descriptives of all study variables). 185 of the 245 participating women (76%) were post-menopausal. No participant reported ever using hormonal contraception or dietary supplements with consistency. No Tsimane was excluded based on any health condition that can affect BMD or fracture risk.

Comparative American BMD data were collected among asymptomatic subjects from greater Los Angeles as part of a different study (described in *Budoff et al., 2010*). Briefly, 9585 subjects (43% female; mean age = 56) underwent coronary artery calcification (CAC) scanning for evaluation of subclinical atherosclerosis, after exclusion of participants with vertebral deformities or fractures. Subjects had no known bone disease (see *Appendix 1—table 7* for additional details). Two American data sources are used to compare Tsimane and American fracture prevalence: a subset from the MESA study (*Budoff et al., 2011*) and a subset reported in *Budoff et al. (2013)* (see *Appendix 1—table 8* for additional details); data from these American subsets were matched to Tsimane by age, sex and weight (±5 kg), and then merged to create a single American comparison sample.

Institutional IRB approval was granted by UNM (HRRC # 07–157) and UCSB (# 3-16-0766) for the Tsimane research, as was informed consent at three levels: (1) Tsimane government that oversees research projects, (2) village leadership and (3) study participants.

### Thoracic computed tomography (CT)

Tsimane CT scans were conducted at the Hospital Presidente German Busch in Trinidad, Bolivia using a 16-detector row scanner (GE Brightspeed, Milwaukee, WI, USA). A licensed radiology technician acquired a single, ECG-gated non-contrast thoracic scan as part of a broader project on atherosclerosis, including CAC assessment (see *Kaplan et al., 2017* and Appendix for additional details). Typical multi-detector CT protocols used for evaluation of CAC include imaging the mid-thoracic spine in the reconstructed field of view, facilitating simultaneous evaluation of thoracic vertebrae during a single examination without additional radiation exposure. Tsimane CT settings were: 250 ms exposure, 2.5 mm slice thickness, 0.5 s rotation speed, 120 kVp, and 40 mA with prospective triggering. Refer to the Appendix for details on CT parameters for the comparative American samples; American CT data were all collected at the same institute (Los Angeles Biomedical Research Institute).

## Thoracic vertebral bone mineral density (BMD)

Vertebral BMD was measured manually in each of three consecutive thoracic vertebrae (T7-T10 range) by a radiologist with 20+ years of experience (see Appendix for additional details). BMD measurement started at the level of the section that contained the left main coronary artery (LMCA) caudally (beginning at either T7 or T8, depending on the origin of the LMCA). The center of the region of interest was located at the center of each vertebrae, with a 2–3 mm distance from the cortical shell; this distance ensured that BMD measurements within the vertebral body excluded the cortical bone of the vertebral shell. For each vertebrae, the radiologist manually positioned a circular region of interest while demarcating cortical from trabecular bone based on visual inspection. Any area with large vessels, bone island fractures and calcified herniated disks were excluded as much as possible from the region of interest with use of the manual free tracing protocol. Mean BMD for the three consecutive thoracic vertebrae was then calculated. This BMD measure is strongly positively correlated (Pearson $r$'s > 0.9) with lumbar vertebral BMD (*Budoff et al., 2012*). CT-derived BMD estimates can be obtained with and without calibration phantoms. Phantomless BMD estimates correlate strongly (Pearson $r = 0.99$) with standard phantom-based CT BMD estimates (*Budoff et al., 2013*). Hounsfield units were converted to BMD (mg/cm$^3$) using a calibration phantom of known density or a scanner-specific mean calibration factor for the T7-T10 vertebrae from scans performed without the phantom. All BMD measurements used in this study were performed at the Los Angeles Biomedical Research Institute.

## Thoracic vertebral fracture

For each subject the radiologist classified seven vertebrae (T6-T12) according to Genant's semiquantitative technique (GST) (*Genant et al., 1993*). While there is no consensus regarding the radiologic definition of vertebral fracture, the GST provides highly reproducible diagnosis of fractures, is the current clinical technique of choice for diagnosing fracture, and is the most widely used technique for identifying fracture (*Shepherd et al., 2015*). Based on visual inspection, each vertebra is rated according to severity of loss of vertebral height and other qualitative features, including alterations in shape and configuration of the vertebra relative to adjacent vertebrae and expected normal appearances. Each vertebra is classified into one of five categories: normal (grade 0); mild fracture (grade 1; approximately 20–25% reduction in anterior, middle, and/or posterior vertebral height, and a 10–20% reduction in projected vertebral area); moderate fracture (grade 2; 25–40% reduction in any height and a 20–40% reduction in area); and severe fracture (grade 3; >40% reduction in any height and area). A grade 0.5 indicates borderline deformed vertebra (<20% reduction in any height) that is not considered to be a definitive fracture (see Appendix for additional details). Each subject is assigned one grade representing a summary measure of all seven vertebrae. Subjects with >1 vertebral deformity are classified according to their most severe deformity. Subjects are considered to present vertebral fracture if any vertebral body is graded at least mildly deformed (i.e. grade ≥1); subjects are considered to present no fracture if graded 0 or 0.5. Given recent analyses (*Lentle et al., 2018*) showing lower observer agreement for mild fractures (the most common) relative to moderate and severe fractures, we repeat analyses using a more conservative fracture definition (i.e. grade ≥2). All fracture measurements used in this study were performed at the Los Angeles Biomedical Research Institute.

## Socio-demographics and anthropometrics

Individuals for whom reliable ages could not be ascertained are not included in analyses (see Appendix for additional details). Reproductive histories were elicited in the Tsimane language. IBI refers to the number of months between live births for women with ≥2 live births. Self-reported ages at menarche and menopause were recorded during medical exams conducted by physicians of the Tsimane Health and Life History Project (THLHP). During annual THLHP census updates we also coded for each participant their village of residence (from which we derived via GPS residential proximity to the closest market town of San Borja), self-reported Spanish fluency (0 = none; 1 = moderate; 2 = fluent) and schooling (# years) as indicators of modernization.

Height and weight were measured during THLHP medical exams using a Seca stadiometer (Road Rod) and Tanita scale (BC-1500). The scale uses a method of bioelectrical impedance analysis to

estimate percent body fat. Using weight and percent body fat we calculated fat mass (weight*percent body fat) and fat-free mass (weight – fat mass).

## Data analysis

The two outcome variables indicating bone strength are thoracic vertebral BMD and thoracic vertebral fracture. Sexes are analyzed separately given the sex-specific nature of hypotheses and to minimize confounding by unobserved factors. General linear models are used to test for associations between Tsimane women's BMD and reproductive effort (see Appendix for additional details). We compared BMD of Tsimane and age- and sex-matched Americans using age-standardized means. We use a parametric test (one sample *t* test) to evaluate whether population-level differences in mean BMD within each decade are significant (p<0.05), specifying as the test value the US means published in *Budoff et al. (2010)*. We compare Tsimane and American fracture prevalence using age-standardized values, and using log-binomial generalized linear models (GLMs). GLMs are used to test for effects of BMD and women's reproductive effort on the probability of fracture. Both continuous and categorical (e.g. median split) measures of reproductive effort are used to analyze their associations with BMD and fracture risk. Unless otherwise noted, anthropometric and socio-demographic covariates are included in a stepwise fashion in regressions (see Appendix for descriptive analyses of Tsimane BMD by age and sex, and by anthropometrics [*Appendix 1—tables 19–20*]). Participants with any missing values are removed from analyses.

## Acknowledgements

We thank the Tsimane for participating and THLHP personnel for collecting and coding data. Jay Stock and two anonymous reviewers provided detailed comments that improved the quality of this manuscript. Funding was provided by the National Institutes of Health/National Institute on Aging (R01AG024119), the Center for Evolutionary Medicine at Arizona State University, and the University of California-Santa Barbara Academic Senate. JS acknowledges IAST funding from the French National Research Agency (ANR) under the Investments for the Future (Investissements d'Avenir) program, grant ANR-17-EURE-0010.

## Additional information

### Funding

| Funder | Grant reference number | Author |
|---|---|---|
| National Institutes of Health | R01AG024119 | Jonathan Stieglitz<br>Benjamin C Trumble<br>Caleb E Finch<br>Hillard Kaplan<br>Michael D Gurven |
| Arizona State University | | Benjamin C Trumble |
| University of California, Santa Barbara | | Michael D Gurven |
| Agence Nationale de la Recherche | ANR-17-EURE-0010 | Jonathan Stieglitz |

The funders had no role in study design, data collection and interpretation, or the decision to submit the work for publication.

### Author contributions

Jonathan Stieglitz, Conceptualization, Resources, Data curation, Software, Formal analysis, Supervision, Funding acquisition, Validation, Investigation, Visualization, Methodology, Writing—original draft, Project administration, Writing—review and editing; Benjamin C Trumble, Funding acquisition, Methodology, Supervision, Project administration, Writing—review and editing; HORUS Study Team, Supervision, Validation, Investigation, Methodology; Caleb E Finch, Project administration, Writing—review and editing; Dong Li, Data curation, Investigation, Methodology;

Matthew J Budoff, Resources, Supervision, Methodology, Project administration; Hillard Kaplan, Conceptualization, Resources, Supervision, Funding acquisition, Investigation, Methodology, Project administration; Michael D Gurven, Funding acquisition, Investigation, Supervision, Project administration, Writing—review and editing

### Author ORCIDs
Jonathan Stieglitz (iD) https://orcid.org/0000-0001-5985-9643
Caleb E Finch (iD) https://orcid.org/0000-0002-7617-3958

### Ethics
Human subjects: Institutional IRB approval was granted by UNM (HRRC # 07-157) and UCSB (# 3-16-0766), as was informed consent at three levels: (1) Tsimane government that oversees research projects, (2) village leadership and (3) study participants.

### Decision letter and Author response
Decision letter https://doi.org/10.7554/eLife.48607.033
Author response https://doi.org/10.7554/eLife.48607.034

## Additional files
### Supplementary files
• Transparent reporting form
DOI: https://doi.org/10.7554/eLife.48607.006

### Data availability
The data that support the findings of this study are available on Dryad (https://doi.org/10.5061/dryad.rf0g0md).

The following dataset was generated:

| Author(s) | Year | Dataset title | Dataset URL | Database and Identifier |
|---|---|---|---|---|
| Stieglitz J, Trumble B, Team HS, Finch C, Li D, Budoff M, Kaplan H, Gurven M | 2019 | Computed tomographyshows high fracture prevalence among physically activeforager-horticulturalists with high fertility | https://doi.org/10.5061/dryad.rf0g0md | Dryad DigitalRepository, 10.5061/dryad.rf0g0md |

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

## Appendix

DOI: https://doi.org/10.7554/eLife.48607.007

### Additional results

#### Tsimane women's thoracic vertebral BMD declines with early age at first birth and short IBI

**Appendix 1—table 1.** General linear model: effect of age at first birth on thoracic vertebral BMD for Tsimane women.

| Parameter | Std. β | p |
|---|---|---|
| Age at 1st birth (years, logged[a]) | 0.099 | 0.036 |
| Age (years) | −0.665 | <0.001 |
| Fat-free mass (kg) | 0.106 | 0.031 |
| Adjusted $R^2$ | 0.508 | |
| N | 213 | |

[a]Age at first birth is not normally distributed and is thus log-transformed (Std. $\beta_{\text{Age at 1st birth [years]}}$=0.110, p=0.019, controlling for age and fat-free mass).

DOI: https://doi.org/10.7554/eLife.48607.008

**Appendix 1—table 2.** General linear models: effects of parity and age at first birth on thoracic vertebral BMD for Tsimane women. Continuous and categorical parity measures (models 1–2, respectively) were added to the baseline model shown in *Appendix 1—table 1*.

| Parameter | Model 1: continuous parity measure | | Model 2: categorical parity measure | |
|---|---|---|---|---|
| | Std. β | p | Std. β | p |
| # births[a] | −0.069 | 0.216 | —— | —— |
| >9 births (vs. ≤ 9 births[b]) | —— | —— | −0.123 | 0.219 |
| Age at 1st birth (years, logged) | 0.075 | 0.145 | 0.081 | 0.105 |
| Age (years) | −0.656 | <0.001 | −0.655 | <0.001 |
| Fat-free mass (kg) | 0.108 | 0.028 | 0.114 | 0.021 |
| Adjusted $R^2$ | 0.509 | | 0.509 | |
| N | 213 | | 213 | |

[a]Logging # births does not substantively affect results.
[b]Substituting other categorical measures of parity (e.g. quartiles) does not substantively affect results.

DOI: https://doi.org/10.7554/eLife.48607.009

**Appendix 1—table 3.** General linear models: effects of mean IBI, parity and age at first birth on thoracic vertebral BMD for Tsimane women. Continuous and categorical mean IBI measures (models 1–2 and models 3–4, respectively) were added to the models presented in *Appendix 1—table 2*. Variance inflation factors (all <1.6) do not indicate a high degree of multicollinearity.

| Parameter | Model 1 | | Model 2 | | Model 3 | | Model 4 | |
|---|---|---|---|---|---|---|---|---|
| | Std. β | p | Std. β | p | Std. β | p | Std. β | p |
| Mean IBI (months[a]) | 0.074 | 0.246 | 0.074 | 0.185 | —— | —— | —— | —— |
| Short mean IBI (<29.7 months; vs. ≥29.7[b]) | —— | —— | —— | —— | −0.185 | 0.084 | −0.184 | 0.075 |
| # births | −0.025 | 0.717 | —— | —— | −0.019 | 0.774 | —— | —— |
| >9 births (vs. ≤ 9 births) | —— | —— | −0.068 | 0.528 | —— | —— | −0.041 | 0.712 |
| Age at 1st birth (years, logged) | 0.090 | 0.092 | 0.089 | 0.076 | 0.102 | 0.060 | 0.102 | 0.046 |
| Age (years) | −0.664 | <0.001 | −0.661 | <0.001 | −0.661 | <0.001 | −0.659 | <0.001 |
| Fat-free mass (kg) | 0.107 | 0.029 | 0.111 | 0.025 | 0.106 | 0.031 | 0.108 | 0.029 |
| Adjusted R² | 0.510 | | 0.511 | | 0.514 | | 0.514 | |
| N | 212 | | 212 | | 212 | | 212 | |

aLogging mean IBI does not substantively affect results.
bMean IBI is not calculated for parous women with only one birth (n = 1), hence the reduction in sample size relative to *Appendix 1—tables 1–2*.

DOI: https://doi.org/10.7554/eLife.48607.010

**Appendix 1—table 4.** General linear model: reduced form including only significant (p<0.05) predictors of Tsimane women's thoracic vertebral BMD.

| Parameter | Std. β | p |
|---|---|---|
| Short mean IBI (<29.7 months; vs. ≥ 29.7) | −0.201 | 0.032 |
| Age at 1st birth (years, logged) | 0.109 | 0.022 |
| Age (years) | −0.663 | <0.001 |
| Fat-free mass (kg) | 0.105 | 0.031 |
| Adjusted R² | 0.517 | |
| N | 212 | |

DOI: https://doi.org/10.7554/eLife.48607.011

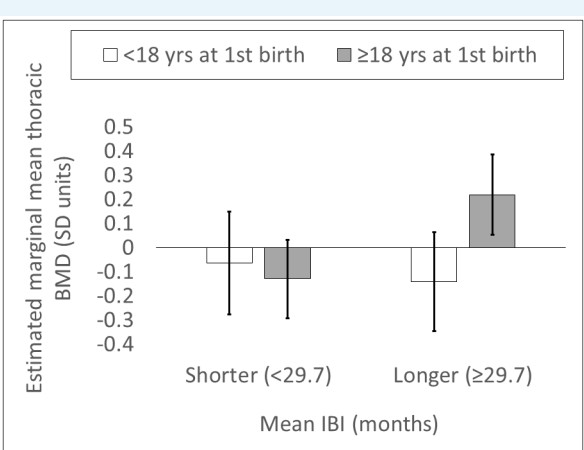

**Appendix 1—figure 1.** Estimated marginal mean thoracic vertebral BMD (SD units; 95% CI) by mean IBI and age at first birth (n = 212 Tsimane women). Values are obtained from a general linear model including the following parameters: mean IBI <29.7 months dummy, age at first birth <18 years dummy, mean IBI <29.7 months dummy*age at first birth <18 years dummy, age, and fat-free mass. Mean IBI <29.7 months dummy*age at first birth <18 years dummy interaction p=0.027. Controls (age and fat-free mass) are held at sample means.
DOI: https://doi.org/10.7554/eLife.48607.012

**Appendix 1—table 5.** General linear models: reduced model shown in *Appendix 1—table 4* also controlling for indicators of modernization, that is, residential proximity to the closest market town of San Borja (models 1–3), Spanish fluency (model 2) and schooling (model 3).

| Parameter | Model 1 Std. β | p | Model 2 Std. β | p | Model 3 Std. β | p |
|---|---|---|---|---|---|---|
| Short mean IBI (<29.7 months; vs. ≥ 29.7) | −0.208 | 0.026 | −0.222 | 0.020 | −0.213 | 0.026 |
| Age at 1st birth (years, logged) | 0.124 | 0.010 | 0.141 | 0.004 | 0.139 | 0.005 |
| Age (years) | −0.665 | <0.001 | −0.673 | <0.001 | −0.675 | <0.001 |
| Fat-free mass (kg) | 0.105 | 0.031 | 0.112 | 0.023 | 0.113 | 0.022 |
| Close residential proximity to market town (<34.7 km; vs. ≥34.7[a]) | 0.165 | 0.080 | 0.147 | 0.126 | 0.157 | 0.102 |
| Spanish fluency (moderate or high vs. none[b]) | ——— | ——— | −0.006 | 0.954 | ——— | ——— |
| Schooling (any vs. none[b]) | ——— | ——— | ——— | ——— | 0.022 | 0.853 |
| Adjusted R² | 0.521 | | 0.521 | | 0.522 | |
| N | 212 | | 207 | | 207 | |

[a]Median split.
[b]Data are missing for five women.

DOI: https://doi.org/10.7554/eLife.48607.013

**Appendix 1—table 6.** General linear models: reduced model shown in *Appendix 1—table 4* also controlling for residential proximity to the closest market town of San Borja (models 1–2), young age at menarche (model 1) and young age at menopause (model 2).

| Parameter | Model 1 | | Model 2 | |
|---|---|---|---|---|
| | Std. β | p | Std. β | p |
| Short mean IBI (<29.7 months; vs. ≥ 29.7) | −0.177 | 0.068 | −0.250 | 0.020 |
| Age at 1st birth (years, logged) | 0.132 | 0.008 | 0.163 | 0.002 |
| Age (years) | −0.659 | <0.001 | −0.676 | <0.001 |
| Fat-free mass (kg) | 0.133 | 0.009 | 0.136 | 0.017 |
| Close residential proximity to market town (<34.7 km; vs. ≥ 34.7) | 0.112 | 0.253 | 0.078 | 0.464 |
| Young age at menarche (<13.3 years vs. ≥ 13.3 [a,b]) | 0.008 | 0.933 | —— | —— |
| Young age at menopause (<50 years vs. ≥ 50 [a,c]) | —— | —— | −0.176 | 0.120 |
| Adjusted R² | 0.543 | | 0.559 | |
| N | 188 | | 152 | |

[a]Median split.
[b]Data are missing for 24 women.
[c]Some women either haven't yet experienced menopause or cannot recall when they experienced menopause, hence the reduction in sample size for model 2.

DOI: https://doi.org/10.7554/eLife.48607.014

## Thoracic vertebral BMD is lower for Tsimane than Americans, but only for women

**Appendix 1—table 7.** Mean thoracic vertebral BMD (mg/cm$^3$) for Tsimane and Americans[a] by age category and sex.

| Age category (years) | Women | | | Men | | | Total | | |
|---|---|---|---|---|---|---|---|---|---|
| | Tsimane (N) | US[c] (N) | % difference (from US baseline)[d] | Tsimane (N) | US[c] (N) | % difference (from US baseline)[d] | Tsimane (N) | US[c] (N) | % difference (from US baseline)[d] |
| 40–49 | 192.89 (80) | 204.33 (492) | −5.60 (p=0.004) | 192.33 (77) | 194.32 (895) | −1.02 (p=0.547) | 192.62 (157) | 197.87 (1387) | −2.65 (p=0.040) |
| 50–59 | 160.43 (89) | 177.79 (1164) | −9.76 (p<0.001) | 180.06 (98) | 173.13 (1708) | 4.00 (p=0.027) | 170.72 (187) | 175.02 (2872) | −2.46 (p=0.076) |
| 60–69 | 132.79 (49) | 153.90 (1204) | −13.72 (p=0.001) | 150.26 (56) | 158.46 (1422) | −5.18 (p=0.069) | 142.11 (105) | 156.37 (2626) | −9.12 (p<0.001) |
| ≥70[b] | 103.29 (27) | 127.14 (1048) | −18.76 (p<0.001) | 135.82 (31) | 139.27 (1097) | −2.48 (p=0.543) | 120.68 (58) | 133.34 (2145) | −9.50 (p=0.005) |
| Total[e] | 163.85 (245) | 179.91 (3908) | −8.93 (———) | 176.21 (262) | 176.64 (5122) | −0.24 (———) | 170.37 (507) | 177.61 (9030) | −4.08 (———) |

[a]US data represent asymptomatic subjects from greater Los Angeles (described in **Budoff et al., 2010**). Briefly, 9585 subjects (43% female; mean age = 56) underwent coronary artery calcification (CAC) scanning for evaluation of subclinical atherosclerosis, after exclusion of participants with vertebral deformities or fractures. Subjects had no known bone disease.

[b]Maximum age for Tsimane women and men = 91 and 94 years, respectively (US maximum age = 90 for both sexes).

[c]US means are weighted by sample sizes (shown in Table 1 of **Budoff et al., 2010**). This table reports age sub-groups in two-year intervals (e.g. 41–42, 43–44, etc.); for two-year intervals spanning multiple decades that overlap with the age categories shown in this table (i.e. 39–40, 49–50, 59–60 and 69–70), we assume that each year of the two-year interval contributes 50% of the sample.

[d]P-value from a one-sample t test, including as the test value the weighted mean from **Budoff et al. (2010)**.

[e]Age-standardized means are shown to account for differences in age distributions across populations. We use the Tsimane adult age distribution (calculated from the 2015 THLHP census) as the standard. To calculate age-standardized means, means for each age category and population (i.e. unadjusted means for Tsimane and weighted means for US) are multiplied by the proportional representation of that age category in the 2015 THLHP census, and then summed across all age categories.

DOI: https://doi.org/10.7554/eLife.48607.015

## Thoracic vertebral fracture prevalence is higher for Tsimane than Americans

**Appendix 1—table 8.** Age-specific thoracic vertebral (T6-T12) fracture prevalence (% with fracture grade $\geq$1) for Tsimane and Americans[a] by sex[b].

| Age category (years) | Women (n = 491) | | Men (n = 524) | | Total (n = 1,015) | |
|---|---|---|---|---|---|---|
| | Tsimane (95% CI) | US (95% CI) | Tsimane (95% CI) | US (95% CI) | Tsimane (95% CI) | US (95% CI) |
| 40–49 | 14 (6–21) | 4 (<1–8) | 34 (23–45) | 12 (4–19) | 24 (17–30) | 8 (3–12) |
| 50–59 | 25 (16–34) | 15 (7–23) | 37 (27–46) | 9 (4–15) | 31 (24–38) | 12 (7–16) |
| 60–69 | 20 (9–32) | 13 (4–22) | 36 (23–49) | 11 (2–20) | 29 (20–37) | 12 (6–18) |
| $\geq$70[c] | 15 (<1–29) | 9 (<1–21) | 43 (25–62) | 16 (2–29) | 30 (18–42) | 13 (4–22) |
| Total (crude) | 19 (14–24) | 11 (7–14) | 36 (31–42) | 11 (7–15) | 28 (24–32) | 11 (8–14) |
| Total (age-standardized[d]) | 18 | 9 | 36 | 11 | 27 | 10 |

[a]US data are from two sources: 1) a subset of MESA study participants (described in **Budoff et al., 2011**), and 2) a subset of study participants in the greater Los Angeles area (described in **Budoff et al., 2013**). Briefly, the MESA cohort is a longitudinal, population-based study of 6814 adults (54% female; mean age = 62) free of clinical cardiovascular disease, representing six areas in the US: Baltimore, MD, Chicago, IL, Forsyth County, NC, Los Angeles, CA, New York, NY, and St. Paul, MN. Regarding participants from the greater Los Angeles area, data were collected among 4126 asymptomatic subjects (51% female; mean age = 64) who underwent CAC scanning for evaluation of subclinical atherosclerosis. MESA and Los Angeles-only subsets were first matched for age and sex with the Tsimane sample, and then merged to create a single US comparison sample.

[b]For women (n = 245 Tsimane and 246 US), sample sizes for ages 40–49 are 80 Tsimane and 77 US, for ages 50–59 89 Tsimane and 86 US, for ages 60–69 49 Tsimane and 60 US, and for ages 70+ 27 Tsimane and 23 US. For men (n = 261 Tsimane and 263 US), sample sizes for ages 40–49 are 77 Tsimane and 69 US, for ages 50–59 98 Tsimane and 108 US, for ages 60–69 56 Tsimane and 54 US, and for ages 70+ 30 Tsimane and 32 US.

[c]Maximum age for Tsimane women and men = 91 and 94 years, respectively. Maximum age for US women and men = 92 and 91 years, respectively.

[d]Age-standardized prevalences are shown to account for differences in age distributions across populations. We use the Tsimane adult age distribution (calculated from the 2015 THLHP census) as the standard. Prevalence for each age category and population is multiplied by the proportional representation of that age category in the 2015 THLHP census, and then summed across all age categories.

DOI: https://doi.org/10.7554/eLife.48607.016

**Appendix 1—table 9.** Age-specific thoracic vertebral (T6-T12) fracture prevalence (% with fracture grade ≥2) for Tsimane and Americans by sex.

| Age category (years) | Women (n = 491) | | Men (n = 524) | | Total (n = 1,015) | |
|---|---|---|---|---|---|---|
| | Tsimane (95% CI) | US (95% CI) | Tsimane (95% CI) | US (95% CI) | Tsimane (95% CI) | US (95% CI) |
| 40–49 | 5 (<1–10) | 0 (–) | 5 (<1–10) | 1 (<1–4) | 5 (2–9) | 1 (<1–2) |
| 50–59 | 8 (2–14) | 3 (<1–7) | 14 (7–21) | 2 (<1–4) | 11 (7–16) | 3 (<1–5) |
| 60–69 | 4 (<1–10) | 3 (<1–8) | 9 (1–17) | 7 (<1–15) | 7 (2–12) | 5 (1–9) |
| ≥70 | 11 (<1–24) | 4 (<1–13) | 23 (7–39) | 3 (<1–9) | 18 (7–28) | 4 (<1–9) |
| Total (crude) | 7 (3–10) | 2 (<1–4) | 11 (8–15) | 3 (1–5) | 9 (7–12) | 3 (1–4) |
| Total (age-standardized) | 6 | 2 | 10 | 2 | 8 | 3 |

DOI: https://doi.org/10.7554/eLife.48607.017

**Appendix 1—table 10.** Log-binomial generalized linear models: women's adjusted relative risk of fracture (95% CI) by population and fracture grade controlling for age (n=491).

| Parameter | Model one outcome: grade 0.5 fracture (borderline) | Model two outcome: grade one fracture (mild) | Model three outcome: grade two fracture (moderate) | Model four outcome: grade three fracture (severe) |
|---|---|---|---|---|
| Tsimane (vs. American) | 1.14 (0.80–1.64) | 1.56¶ (0.91–2.65) | 3.01* (1.11–8.17) | 1.33 (0.09–18.92) |
| Age (per SD increase) | 1.04 (0.87–1.23) | 1.05 (0.82–1.34) | 1.02 (0.68–1.55) | 6.26** (1.46–26.79) |

**p≤0.01 *p≤0.05 ¶p≤0.1

DOI: https://doi.org/10.7554/eLife.48607.018

**Appendix 1—table 11.** Log-binomial generalized linear models: men's adjusted relative risk of thoracic vertebral (T6-T12) fracture (95% CI) by population and fracture grade controlling for age (n = 524).

| Parameter | Model one outcome: grade 0.5 fracture (borderline) | Model two outcome: grade one fracture (mild) | Model three outcome: grade two fracture (moderate) | Model four outcome: grade three fracture (severe) |
|---|---|---|---|---|
| Tsimane (vs. American) | 2.27*** (1.58–3.27) | 3.12*** (1.97–4.95) | 4.70*** (1.98–11.17) | 1.48 (0.21–10.63) |
| Age (per SD increase) | 0.88 (0.74–1.04) | 0.99 (0.81–1.19) | 1.16 (0.86–1.56) | 4.79*** (1.94–11.81) |

***p≤0.001 **p≤0.01 *p≤0.05 ¶p≤0.1

DOI: https://doi.org/10.7554/eLife.48607.019

**Lower thoracic vertebral BMD increases risk of thoracic vertebral fracture among Tsimane, particularly for women. Short IBI additionally increases Tsimane women's fracture risk**

Appendix 1—table 12. Sample characteristics for Tsimane women (n = 245) with and without any thoracic vertebral (T6-T12) fracture (grade $\geq$1).

| Variable | A) Fracture (n=47[a]) | | B) No fracture (n=198[a]) | | A vs. B[b] |
|---|---|---|---|---|---|
| | Mean | SE | Mean | SE | |
| Thoracic vertebral BMD (mg/cm$^3$) | 142.66 | 5.93 | 163.13 | 3.20 | p=0.006 |
| Age (years) | 56.67 | 1.45 | 56.09 | 0.73 | p=0.490 |
| Height (cm) | 149.79 | 0.63 | 150.47 | 0.38 | p=0.631 |
| Weight (kg) | 56.07 | 1.71 | 54.40 | 0.68 | p=0.404 |
| BMI (kg/m$^2$) | 24.94 | 0.72 | 23.99 | 0.27 | p=0.355 |
| Body fat (%) | 26.83 | 1.34 | 25.49 | 0.55 | p=0.344 |
| Fat mass (kg) | 15.72 | 1.23 | 14.37 | 0.48 | p=0.403 |
| Fat-free mass (kg) | 40.14 | 0.93 | 40.03 | 0.37 | p=0.930 |
| Age at menarche (years) | 13.21 | 0.09 | 13.32 | 0.04 | p=0.352 |
| Age at menopause (years) | 49.76 | 0.64 | 49.16 | 0.30 | p=0.408 |
| Age at first birth (years) | 18.49 | 0.67 | 18.91 | 0.28 | p=0.427 |
| Parity (# births) | 9.72 | 0.54 | 8.89 | 0.22 | p=0.072 |
| Mean inter-birth interval (months) | 30.69 | 2.61 | 34.34 | 1.17 | p=0.010 |
| Residential proximity to market (km) | 49.01 | 5.28 | 50.63 | 2.71 | p=0.955 |
| Spanish fluency (1 = fluent/ moderate; 0 = none) | 0.24 | 0.07 | 0.37 | 0.04 | p=0.114 |
| Schooling (# years) | 0.36 | 0.15 | 0.59 | 0.10 | p=0.318 |

[a]Represents maximum possible sample size (i.e. no missing data).
[b]P-value from Mann-Whitney U or $\chi^2$ test.

DOI: https://doi.org/10.7554/eLife.48607.020

**Appendix 1—table 13.** Log-binomial generalized linear models: effects of thoracic vertebral BMD (models 1–2) and mean IBI (model 2) on the probability of thoracic vertebral (T6-T12) fracture (grade ≥1) for Tsimane women. Relative risk (95% CI) is shown per SD increase.

| Parameter | Model 1: controlling for age and anthropometrics | | | Model 2: + mean IBI | | |
|---|---|---|---|---|---|---|
| | Exp(β) | 95% CI | p | Exp(β) | 95% CI | p |
| Thoracic vertebral BMD (mg/cm$^3$) | 0.542 | 0.352–0.837 | 0.006 | 0.540 | 0.345–0.845 | 0.007 |
| Mean IBI (months) | ——— | ——— | ——— | 0.379 | 0.165–0.866 | 0.021 |
| Age (years) | 0.676 | 0.449–1.017 | 0.061 | 0.640 | 0.423–0.968 | 0.034 |
| Height (cm) | 0.868 | 0.660–1.141 | 0.310 | 0.881 | 0.662–1.172 | 0.385 |
| Fat mass (kg) | 1.278 | 0.978–1.670 | 0.073 | 1.291 | 1.006–1.658 | 0.045 |
| N | 219 | | | 212 | | |

DOI: https://doi.org/10.7554/eLife.48607.021

**Appendix 1—table 14.** Log-binomial generalized linear models: effects of thoracic vertebral BMD and mean IBI on the probability of thoracic vertebral (T6-T12) fracture for Tsimane women (n = 212) by fracture grade. Grade 3 (severe) fracture risk is not modeled because no grade three cases are present for the sample of women with complete data.
Relative risk (95% CI) is shown per SD increase.

| Parameter | Model one outcome: grade 0.5 fracture (borderline) | Model two outcome: grade one fracture (mild) | Model three outcome: grade two fracture (moderate) |
|---|---|---|---|
| Thoracic vertebral BMD (mg/cm$^3$) | 0.92 (0.62–1.36) | 0.73 (0.43–1.24) | 0.27** (0.10–0.71) |
| Mean IBI (months) | 1.02 (0.81–1.29) | 0.33* (0.12–0.93) | 0.49 (0.09–2.72) |
| Age (years) | 1.06 (0.76–1.49) | 0.85 (0.51–1.41) | 0.34** (0.16–0.72) |
| Height (cm) | 1.01 (0.76–1.34) | 1.00 (0.71–1.40) | 0.63 (0.32–1.25) |
| Fat mass (kg) | 0.77¶ (0.57–1.05) | 1.26¶ (0.97–1.63) | 1.25 (0.64–2.45) |

**p≤0.01 *p≤0.05 ¶p≤0.1

DOI: https://doi.org/10.7554/eLife.48607.022

**Appendix 1—table 15.** Sample characteristics for Tsimane men (n = 261) with and without any thoracic vertebral (T6-T12) fracture (grade $\geq$1).

| Variable | A) Fracture (n=95[a]) | | B) No fracture (n=166[a]) | | A vs. B[b] |
|---|---|---|---|---|---|
| | Mean | SE | Mean | SE | |
| Thoracic vertebral BMD (mg/cm$^3$) | 169.61 | 3.84 | 173.64 | 2.80 | p=0.454 |
| Age (years) | 57.04 | 1.07 | 55.74 | 0.71 | p=0.392 |
| Height (cm) | 160.60 | 0.53 | 161.86 | 0.43 | p=0.034 |
| Weight (kg) | 63.33 | 0.94 | 61.96 | 0.57 | p=0.303 |
| BMI (kg/m$^2$) | 24.49 | 0.29 | 23.63 | 0.19 | p=0.014 |
| Body fat (%) | 19.05 | 0.76 | 16.80 | 0.46 | p=0.026 |
| Fat mass (kg) | 12.46 | 0.63 | 10.56 | 0.35 | p=0.034 |
| Fat-free mass (kg) | 51.22 | 0.68 | 51.39 | 0.47 | p=0.936 |
| Residential proximity to market (km) | 51.28 | 3.80 | 56.60 | 3.07 | p=0.315 |
| Spanish fluency (1 = fluent/ moderate; 0 = none) | 0.77 | 0.05 | 0.82 | 0.03 | p=0.404 |
| Schooling (# years) | 2.02 | 0.38 | 1.70 | 0.24 | p=0.639 |

[a]Represents maximum possible sample size (i.e. no missing data).
[b]P-value from Mann-Whitney U or $\chi^2$ test.

DOI: https://doi.org/10.7554/eLife.48607.023

**Appendix 1—table 16.** Log-binomial generalized linear model: effect of thoracic vertebral BMD on the probability of thoracic vertebral (T6-T12) fracture (grade ≥1) for Tsimane men (n=227[a]). Relative risk (95% CI) is shown per SD increase.

| Parameter | Exp(β) | 95% CI | p |
|---|---|---|---|
| Thoracic vertebral BMD (mg/cm$^3$) | 1.046 | 0.862–1.269 | 0.650 |
| Age (years) | 1.010 | 0.822–1.241 | 0.924 |
| Height (cm) | 0.822 | 0.701–0.963 | 0.015 |
| Fat mass (kg) | 1.366 | 1.176–1.587 | <0.001 |

[a]Fracture data are missing for one man with BMD and anthropometric data. We also omitted a man whose height (138.2 cm) was 4.3 SDs below the mean.

DOI: https://doi.org/10.7554/eLife.48607.024

**Appendix 1—table 17.** Log-binomial generalized linear models: effect of thoracic vertebral BMD on the probability of thoracic vertebral (T6-T12) fracture for Tsimane men by fracture grade. Relative risk (95% CI) is shown per SD increase.

| Parameter | Model one outcome: grade 0.5 fracture (borderline) | Model two outcome: grade one fracture (mild) | Model three outcome: grade two fracture (moderate) | Model four outcome: grade three fracture (severe) |
|---|---|---|---|---|
| Thoracic vertebral BMD (mg/cm$^3$) | 0.76*[a] (0.60–0.96) | 0.96 (0.73–1.26) | 1.24[a] (0.83–1.84) | 0.49 (0.02–11.09) |
| Age (years) | 0.81¶[a] (0.64–1.03) | 0.82 (0.61–1.10) | 1.40¶[a] (0.93–2.12) | 2.87 (0.24–34.91) |
| Height (cm) | 1.18¶[b] (0.98–1.42) | 0.75** (0.60–0.93) | 0.94[c] (0.65–1.37) | 0.70 (0.11–4.60) |
| Fat mass (kg) | 0.80*[a] (0.65–0.995) | 1.41*** (1.15–1.73) | 1.26[a] (0.90–1.78) | 0.81 (0.09–7.46) |
| N | 227[a] | 227 | 227[a] | 227 |

***p≤0.001 **p≤0.01 *p≤0.05 ¶p≤0.1
[a]Model parameters include BMD, age and fat mass (not height). Joint inclusion of both height and fat mass yields invalid estimates.
[b]Model parameters include BMD (adjusted RR = 0.75, 95% CI: 0.61–0.92, p=0.006), age (adjusted RR = 0.84, 95% CI: 0.66–1.07, p=0.159) and height (not fat mass, n = 253).
[c]Model parameters include BMD (adjusted RR = 1.21, 95% CI: 0.81–1.82, p=0.358), age (adjusted RR = 1.40, 95% CI: 0.92–2.14, p=0.116) and height (not fat mass, n = 253).

DOI: https://doi.org/10.7554/eLife.48607.025

# Additional materials and methods

## Study design and participants

Since 2002 the Tsimane (population ~16,000) have participated in the ongoing Tsimane Health and Life History Project (THLHP; see *Gurven et al., 2017* for a project overview). All Tsimane residing in study villages are eligible to participate, and most choose to do so at least once. Project physicians have conducted annual medical exams on Tsimane of all ages since 2002. A mobile team of physicians, biochemists, and Tsimane research assistants collects data in Tsimane villages on medical and reproductive histories, functional ability, and other aspects of lifestyle (e.g. food production and sharing), in addition to collecting biological specimens (e.g. serum, urine, feces) among a subset.

Between July 2014 and September 2015, men and women aged 40+ years from 59 Tsimane villages were invited to participate in the CT scanning project. At the time, there

were 1214 eligible people living in these villages (the only eligibility criteria were being 40+ years old, self-identifying as Tsimane and willing to participate). 731 individuals were present in their villages at the time and subsequently received a CT scan. Transporting participants from their village to the nearby market town of San Borja was logistically complicated (requiring trekking through the forest, dug-out canoes, rafts propelled by poles pushed off the river bottom, trucks, and cars) and can require up to two days of travel each way. From San Borja to the Beni department capital of Trinidad (where the hospital containing the CT scanner is located) is an additional 6 hr car ride. Due to these logistical complications, participants not in their village at the time we arrived were not sampled. The Tsimane are semi-mobile and often build secondary houses deep in the forest near their horticultural plots, not returning to their village for extended periods of time. Hunting and fishing trips can last days or weeks, and some men engage in wage labor in San Borja or elsewhere (e.g. rural cattle ranches). In an average village, approximately one-third of individuals are away hunting, fishing, working in their horticultural plots, or in San Borja at any given time. Additionally, a major flood in February 2014 resulted in mass migration from some villages, and the creation of several new villages that were not sampled as part of this study, further reducing the number individuals that could be sampled in this study. To address potential sources of sample bias, analyses comparing Tsimane who received CTs and those who did not but who participated in the THLHP's medical exams in Tsimane villages were conducted. There were no significant differences in sex, blood pressure or body fat (see *Kaplan et al., 2017* for additional participant details) and thus the CT sample analyzed here is thought to be representative of all Tsimane aged 40+ years.

Of the 731 Tsimane who received a CT scan, CT data from 507 (69%) were selected with no particular criteria to estimate thoracic vertebral bone mineral density (BMD) and fracture (scans from 224 Tsimane were not assessed due to radiologist time constraints). Among these 507 individuals, some data were missing because of either broken equipment, missing supplies, participant recall problems, absent or sick team personnel who were unable to collect data, or because all relevant thoracic vertebrae did not appear in the CT image field of view (see *Appendix 1—table 18* for sample sizes and descriptives for all study variables).

**Appendix 1—table 18.** Descriptives for all study variables[a].

| Variable | N | Mean | SD | Min | Max |
|---|---|---|---|---|---|
| Thoracic vertebral BMD (mg/cm$^3$) | 507 | 165.85 | 41.27 | 68.91 | 314.99 |
| Any thoracic vertebral (T6-T12) fracture (% grade $\geq$ 1) | 506 | 0.28 | 0.45 | 0.00 | 1.00 |
| Age (years) | 507 | 56.25 | 9.95 | 41.00 | 94.00 |
| Sex (% male) | 507 | 0.52 | 0.50 | 0.00 | 1.00 |
| Height (cm) | 493 | 156.05 | 7.60 | 136.00 | 176.30 |
| Weight (kg) | 493 | 58.70 | 9.73 | 34.60 | 96.90 |
| BMI (kg/m$^2$) | 493 | 24.05 | 3.33 | 16.33 | 38.11 |
| Body fat (%) | 448 | 21.57 | 8.00 | 5.00 | 46.70 |
| Fat mass (kg) | 448 | 12.88 | 6.04 | 1.95 | 42.12 |
| Fat-free mass (kg) | 448 | 45.80 | 7.85 | 27.81 | 73.08 |
| Age at menarche (years) | 214 | 13.30 | 0.56 | 11.75 | 16.29 |
| Age at menopause (years[b]) | 173 | 49.28 | 3.56 | 37.00 | 55.50 |
| Age at first birth (years) | 235 | 18.83 | 3.91 | 12.00 | 37.00 |
| Parity (# live births) | 245 | 9.05 | 3.27 | 0.00 | 17.00 |
| Mean inter-birth interval (months) | 231 | 33.68 | 16.27 | 14.02 | 142.37 |
| Residential proximity to market (km) | 507 | 52.62 | 38.22 | 4.71 | 153.61 |
| Spanish fluency (1 = fluent/ moderate; 0 = none) | 451 | 0.58 | 0.50 | 0.00 | 1.00 |
| Schooling (# years) | 450 | 1.19 | 2.47 | 0.00 | 15.00 |

[a]Data were missing for various reasons (see supplementary text for details).
[b]185 of the 245 participating women (76%) were post-menopausal. 12 women were post-menopausal but could not recall their age at menopause. The remaining 60 women (245-185) were either pre-menopausal (n = 36) or missing data regarding menopausal status (n = 24). Women were categorized as post-menopausal if they reported during THLHP medical exams not having experienced a menstrual cycle in the past year, and were neither pregnant nor lactating at the time of the study.

DOI: https://doi.org/10.7554/eLife.48607.026

# Thoracic computed tomography (CT)

Tsimane CT scans were supervised and reviewed by at least one of the HORUS study team cardiologists and radiologists. Breath-hold instructions were given in the Tsimane language to minimize respiratory motion artifact and misregistration (for additional methodological details see *Kaplan et al., 2017*).

CT scanning procedures used to collect BMD data from greater Los Angeles involved use of electron-beam CT scanners (C-300; GE-Imatron, South San Francisco, California) and a 64–

detector row CT scanner (LightSpeed VCT; GE Medical Systems) at the Los Angeles Biomedical Research Institute. Parameters for electron-beam CT scanning were 130 kVp, 630 mA, and 2.5 or 3 mm slice thickness. The multidetector CT parameters were 120 kVp, 200–600 mA, and 2.5 mm slice thickness. 156 subjects underwent scanning with electron-beam CT and 64–detector row CT on the same day for comparison and normalization of BMDs between scanners. CT scanning to collect MESA sub-sample data, used here to assess fracture prevalence and severity, was performed at the Los Angeles Biomedical Research Institute; procedures involved the use of a C-150 GE-Imatron electron-beam scanner and were otherwise identical to those described in *Budoff et al. (2010)*. Supplementary CT data for assessment of fracture prevalence and severity were also collected at the Los Angeles Biomedical Research Institute using similar parameters.

## Thoracic vertebral bone mineral density (BMD)

CT-measured vertebral BMD is increasingly used for osteoporosis screening because of its ability to provide three-dimensional information compared to traditional dual x-ray absorptiometry two-dimensional images (*Budoff et al., 2012*). The left main coronary artery (LMCA) was set as the reference site to allow reproducible detection of a spinal level for use with cardiac CT scanning. The LMCA is covered in 100% of images and the field of view can be completely reconstructed; therefore, it is an optimal reference point to locate the starting measurement level. The most common location of the LMCA origin is at the lower edge of T7.

Intra-observer variability in BMD measurements using the same measurement technique in a different sample (*Budoff et al., 2010*) was tested on 120 scans by one observer from the Los Angeles Biomedical Research Institute, with 1 week intervals between the two readings. To measure inter-observer variability on 67 randomly selected scans, the results obtained by two radiologists from the Los Angeles Biomedical Research Institute who were blinded to all clinical information and prior measurements were compared using this other sample. Intra- and inter-observer variations in BMD measurements were 2.5% (Bland-Altman plot ratio: 1.00, 95% CI: 0.99–1.00) and 2.6% (Bland-Altman plot ratio: 1.00, 95% CI: 0.99–1.00), respectively.

## Thoracic vertebral fracture

Genant's semi-quantitative technique does not distinguish between wedge (i.e. reduced anterior height), biconcave (i.e. reduced central height) or crush (i.e. reduced posterior height) fractures; most fractures contain a combination of these features and are influenced by the local biomechanics of the spinal level involved. Vertebral fractures are differentiated from other, non-fracture deformities (e.g. osteoarthritis), although these other deformities were not systematically coded.

## Socio-demographics and anthropometrics

Birth years were assigned based on a combination of methods described in detail elsewhere (*Gurven et al., 2007*), including using known ages from written records, relative age lists, dated events, photo comparisons of people with known ages, and cross-validation of information from independent interviews of kin. Each method provides an independent estimate of age, and when estimates yielded a date of birth within a three-year range, the average was generally used.

Outcomes of each pregnancy reported during reproductive histories were recorded as either ending in a live birth or terminating pre-term. Whether miscarriages (including stillbirths) are included or omitted from parity counts does not affect results, and results reported here reflect only live births.

## Data analysis

With respect to general linear models of BMD, variance inflation factors were checked to assess degree of covariance and ensure minimal collinearity. Crude mean BMD values are

used to compare populations for each decade, and age-standardized means are used to compare populations across all decades. We use the Tsimane adult age distribution, estimated from the THLHP 2015 census, as the standard.

**Appendix 1—table 19.** General linear models: anthropometric predictors of thoracic vertebral BMD (mg/cm$^3$) for women. Height is included in models 1 and 3, weight in models 2, 3 and 5, body mass index (BMI) in model 4, body fat (% or fat mass) in models 5–6, and fat-free mass in model 6. Age is controlled in all models. Standardized betas are shown (intercepts omitted).

| Parameter | Model 1 | Model 2 | Model 3 | Model 4 | Model 5 | Model 6 |
|---|---|---|---|---|---|---|
| Height (cm[a]) | 0.040 | ——— | 0.013[c] | ——— | ——— | ——— |
| Weight (kg[a]) | ——— | 0.080¶ | 0.075[c] | ——— | 0.130¶[c] | ——— |
| BMI (kg/m$^{2}$[a]) | ——— | ——— | ——— | 0.070 | ——— | ——— |
| Body fat (%[b]) | ——— | ——— | ——— | ——— | −0.061[c] | ——— |
| Fat mass (kg[b]) | ——— | ——— | ——— | ——— | ——— | 0.021[c] |
| Fat-free mass (kg[b]) | ——— | ——— | ——— | ——— | ——— | 0.087[c] |
| Age (years) | −0.679[***] | −0.670[***] | −0.667[***] | −0.681[***] | −0.661[***] | −0.659[***] |
| Adjusted R$^2$ | 0.484 | 0.489 | 0.487 | 0.488 | 0.498 | 0.498 |
| N | 238 | 238 | 238 | 238 | 219 | 219 |

[***]p≤0.001 **p≤0.01 *p≤0.05 ¶p≤0.1
[a]Data are missing for seven women.
[b]Data are missing for 26 women.
[c]Variance inflation factors do not indicate a high degree of multicollinearity.

DOI: https://doi.org/10.7554/eLife.48607.027

**Appendix 1—table 20.** General linear models: anthropometric predictors of thoracic vertebral BMD (mg/cm$^3$) for men. Height is included in models 1 and 3, weight in models 2, 3 and 5, body mass index (BMI) in model 4, body fat (% or fat mass) in models 5–6, and fat-free mass in model 6. Age is controlled in all models. Standardized betas are shown (intercepts omitted).

| Parameter | Model 1 | Model 2 | Model 3 | Model 4 | Model 5 | Model 6 |
|---|---|---|---|---|---|---|
| Height (cm[a]) | 0.028 | ——— | −0.062[c] | ——— | ——— | ——— |
| Weight (kg[a]) | ——— | 0.137** | 0.170**[c] | ——— | 0.177**[c] | ——— |
| BMI (kg/m$^{2}$[a]) | ——— | ——— | ——— | 0.135** | ——— | ——— |
| Body fat (%[b]) | ——— | ——— | ——— | ——— | −0.072[c] | ——— |
| Fat mass (kg[b]) | ——— | ——— | ——— | ——— | ——— | 0.022[c] |
| Fat-free mass (kg[b]) | ——— | ——— | ——— | ——— | ——— | 0.163**[c] |
| Age (years) | −0.578[***] | −0.556[***] | −0.561[***] | −0.567[***] | −0.508[***] | −0.505[***] |
| Adjusted R$^2$ | 0.307 | 0.325 | 0.325 | 0.324 | 0.300 | 0.302 |
| N | 255 | 255 | 255 | 255 | 229 | 229 |

[***]p≤0.001 **p≤0.01 *p≤0.05 ¶p≤0.1
[a]Data are missing for seven men.
[b]Data are missing for 33 men.
[c]Variance inflation factors do not indicate a high degree of multicollinearity.

DOI: https://doi.org/10.7554/eLife.48607.028

