## [Decision Letter]

Thank you for submitting your article "Computed tomography shows high fracture prevalence among physically active forager-horticulturalists with high fertility" for consideration by *eLife*. Your article has been reviewed by three peer reviewers, and the evaluation has been overseen by Diethard Tautz as the Senior and Reviewing Editor. The following individual involved in review of your submission has agreed to reveal his identity: Jay Stock (Reviewer #1).

The reviewers have discussed the reviews with one another and the Reviewing Editor has drafted this decision to help you prepare a revised submission.

Summary and essential revisions:

Based on the reviewer comments, the discussion of the comments and the editors' assessment, we recommend to revise the manuscript into a version that does not try to test a specific hypothesis, but is more open to different interpretations. The main issue is that the current version favors a life history explanation (which is interesting and compelling), but has not demonstrated that this can be differentiated from other existing explanations. There is already evidence for life history parameters influencing bone mechanics independent of physical activity levels: Evol Med Public Health;2018(1):167-179.

The full comments of the reviewers are attached to this letter. Please respond to them in detail when submitting the revised version.

*Reviewer #1:*

This paper aims to test the hypothesis that modern human skeletal fragility results from greater reproductive effort. Bone mineral density was measured in thoracic vertebrae of Tsimane forager-horticulturalists and compared to American men and women from Los Angeles. No systematic differences were found between Tsimane and American men, but Tsimane women, who have higher rates of fertility, younger ages at first birth, and shorter interbirth intervals, were found to have lower thoracic bone mineral density than American women. Most of the analyses in the paper, however, focus on internal variation within the Tsimane sample, where the results demonstrate that among Tsimane women, earlier age at first birth and shorter interbirth intervals are associated with reduced BMD. Age at menarche and menopause are not associated with BMD, but lower BMD and shorter interbirth intervals increases fracture risk. The results of the analyses of the Tsimane men and women are very interesting and provide an important contribution to the literature, but the most original component of the paper is the interpretation of variation within the Tsimane women and between Tsimane and American women, within the framework of life history theory. In my opinion, Life History Theory is a powerful explanatory model that has application across a wide range of biological phenomenon. The key to evaluating this paper is whether LHT provides a testable alternate hypothesis relative other mechanisms. Does it explain the observed variation or are other mechanisms a parsimonious explanation?

Overall this paper presents an important dataset, and I find the analyses appropriate, however the interpretations need further delimitation with regards to other mechanisms.

With regards to the analyses of variation within the sample of Tsimane women, the authors interpret the results as being related to life history factors: higher fertility, earlier first reproduction, and shorter interbirth intervals. This is an innovative interpretation of intrapopulation variation in bone mass, but to make the argument that life history explains variation requires an assumption that bone has the capacity to recover from mineral losses associated with pregnancy and lactation. The authors cite a few papers in support of the 'full recovery' of BMD following lactation (Prentice, for example), but I am highly skeptical of this literature. In general, trabecular bone lost during adulthood, for any reason, is never fully regained. The studies used to justify this interpretation are dated and use very coarse methods for the characterization of bone quality. I am not convinced that this literature is sufficient to rule out the Tsimane bone loss as being the cumulative effect of multiple periods of gestation and lactation 'raiding' the skeleton for resources. If there was anything less than 100% recovery with each pregnancy, you would find losses in bone as a simple result of maternal provisioning and the direct 'cause' would not necessarily be interbirth intervals, but simply the cumulative bone loss with subsequent pregnancies due to the skeleton being used as a mechanism of resource storage. This is not necessarily incompatible with life history theory, but it is difficult to see how one would differentiate whether resource depletion during pregnancy would be differentiated from other life history parameters in the current dataset. The more pregnancies a woman has, the shorter the interbirth interval and the less bone might recover. In essence, this itself is a 'life history' trait, of maternal sacrifice of bone quality and reinvestment of these resources in offspring, but it doesn't fit the model and interpretation as the authors present it.

The comparison between the Tsimane and American populations is used to draw more general conclusions about variation in humans, and forms the basis of the most significant inferences made in the paper. Here I find the analyses somewhat problematic. The results of these comparisons demonstrate that Tsimane BMD is lower than the Americans, but only for women; and that Tsimane thoracic vertebral fracture prevalence is higher than Americans. These results are interpreted within the context of life-history theory, and it is argued that the lower BMD and higher facture prevalence among the Tsimane women is associated with greater reproductive effort.

The most significant issue is with the characterization of the differences in physical activity levels (PALs) between populations. A few authors have previously discussed broad changes in PAL relative to bone robusticity, but most authors do not use this sort of characterisation. The authors argue that the literature is largely based around the assumption that the transition to agriculture is characterized by a reduction in physical activity levels, and as they consider in the Discussion, this isn't really appropriate. They are correct to point this out in the Discussion, but in setting up the paper around this assumption they are both mischaracterizing most of the literature and setting up a straw man argument. What the literature does suggest is a reduction in terrestrial mobility (lower limb loading) associated with the origins of agriculture in most regions but this is also regionally variable. Change is upper body mechanical loading, however, are much more variable across major dietary transitions, so labour/activity intensity may significantly increase with subsistence transitions in some areas. Virtually no research in the prehistoric biomechanics literature tries to pool upper and lower body (systemic) variation in bone mechanical properties and interpret it in the context of overall PALs. Mischaracterizing the literature this way provides an easy target to knock down. One thing that virtually all biomechanics literature agrees upon is that the mechanical response to loading is variable throughout the body relative to local factors of weight bearing activity and muscle function. Within the context of the current study, it is not clear whether thoracic vertebrae would be expected to track either the mechanics of the lower limb (mobility) or the upper limb (manual labour), or some other more general factors. Many might expect it would track different influences on bone quality such as diet. As the study is pitched, the authors characterize that thoracic BMD should track PALS, and they characterize the literature on terrestrial mobility as being an indicator of PALs. This neither accurately characterizes current knowledge, nor reflects the subtlety of what we do know about bone variation in the past.

A third issue is that of the characterization of the Los Angeles comparative group. The variance in biomechanical properties is usually much higher in contemporary populations on account of the internal genetic, dietary and behavioural diversity. This makes characterization of activity extremely difficult. The authors characterize the 'physical inactivity' hypothesis as predicting that modern women from Los Angeles would have lower BMD, but I'm not sure this is necessarily true. Throughout the body of literature on bone biomechanics within recent human populations, declines in bone strength are not always consistent between hunter-gatherers and agricultural populations, and is highly variable throughout the Holocene. Most notably, contemporary human populations have highly variable skeletal strength properties that are dependent upon individual variation in genetics, diet and activity. The range of variation among contemporary populations is typically so high that one rarely finds statistical differences between modern humans and prehistoric populations that are less variable, as the range of variation in the more homogenous populations is found within contemporary society. Imagine, if you will, trying to characterise the 'behaviour' of a typical person in Los Angeles. Could you do it? For any person chosen at random, there is a chance that they may be virtually sedentary, with extremely low PALs, another may be an ultra-marathon runner with extensive athletic training and competition. As a result, it is generally accepted among bone biomechanics researchers that making general assumptions about activity in contemporary populations is challenging at best. This issue could be addressed with any individual activity data for modern subjects, either by recall or activity trackers. Does any such data exist?

The authors note that Tsimane women have moderate to active PAL levels, while men are highly active. Depending on the model of mechanically induced bone remodeling that one accepts, moderate levels of activity may not be sufficient to engender functional adaptive response in the mechanical properties of bone. When compared to the American sample, the specific PAL and reproductive history of the women is unknown. While it is probably safe to assume that the women did have lower fertility and longer inter-birth intervals, it may not be as safe to assume differences in PALs. The Tsimane women have moderate PALs, the American women may have had highly variable PALs with the same population, and activity levels that may have also been 'moderate'. Very little information is provided here about the American data, either in terms of population characteristics and context, or methods, so the reader cannot evaluate whether the characterization of the American group is appropriate

In terms of methods, the information on how the American data was collected is insufficient as presented. The only details given are – "a matched American sample with directly comparable CT derived indicators of bone strength measured by the same laboratory". Specific details of the equipment and settings are provided for the Tsimane data, were the Los Angeles data collected on the same equipment? With the same settings? Were any objects/participants scanned in each location to validate pooling data from different equipment? This needs to be expanded upon, if only briefly in the text or in supplementary information, so that the reader can evaluate for themselves comparability of approaches.

Overall, while my review has been critical of several aspects of the study and interpretations, I would summarize the work more positively. The Tsimane research has provided an extremely important dataset with very interesting social and biological context. The analyses are appropriate and reveal some very interesting trends in bone mass. My concerns with the paper are primarily about interpretation and assumptions relating to activity levels in the American dataset. Both of these concerns mean it is difficult to evaluate the 'life history' hypothesis relative to other proximate mechanisms, or even more broad variation in genetics, habitual activity (and what influence it may have on thoracic vertebrae), early life nutrition, growth plasticity, prevalence of bottle feeding, or extended lactation that may be associated with bmd losses. It is not clear how life history variation necessarily provides a stronger explanation for the data than would any one of these other factors, so alternate explanations should be considered and set up, where possible, as alternate explanations of the data with different predictions of the underlying variation. I believe this dataset and analysis is important and publishable, but it requires more nuanced hypothesis testing to differentiate life history mechanisms from competing hypotheses.

*Reviewer #2:*

This study investigates a Tsimane population sample and compares it to that in an American cohort with specific questions addressed relating to BMD and fracture risk in thoracic vertebrae of living subjects. Overall, I think this is a robust and timely study that provides alternative explanations for the observed low BMD (or gracilization) in some populations (present and past) beyond the effect of activity levels. It is a well-designed study; the sample sizes are satisfactory, and while there are numerous variables to control for, the authors undertook sufficient analyses. The manuscript is well written and overall was a pleasure to read. The implications of the study are significant in our attempts to understand the observed gracilization in modern humans. One of the highlights of the study is the finding that the Tsimane compared to an American population (sample) show lower BMD especially in women. I also appreciate that the authors outlined the study's limitations that are inherent in this type of study of living subjects. Even so, I have a few comments for the authors that I hope will help improve their manuscript.

1) There are a number of predictions made at the beginning of the manuscript (Introduction) and consequently there are many results to report and discussion to follow. I recommend that the Discussion section follow the order in which the predictions were made so that it is easier for the reader to follow and tie in the predictions and findings.

2) Additionally, in the Introduction there is extensive discussion about gracilization especially in the fossil/archaeological record, however in the Discussion section the subject is a little glossed over. I would suggest that the implications and/or the complexity of interpreting gracilization from a broader perspective be discussed. I realize that the authors do this e.g., in the Discussion, but a little more could be added as a way to explain/justify the value of this study which goes beyond investigating physical activity levels unlike some of the previous work in the fossil and archaeological record.

3) While the authors describe how BMD was measured using the computed tomography, I would suggest that a brief description of whether BMD estimates includes both cortical and trabecular bone and if not, how were the two demarcated? The information regarding how the scanner estimates BMD is found in the supporting information section, if there is room, I would include this in the main Materials and methods section. Also, BMD in three thoracic vertebrae was measured, I suppose the BMD reading is the mean of the three and if so, is this a reading provided automated from the scanner or is this done by the researchers manually after the scans are completed?

4) I found Figure 4 to be a bit confusing to follow and that is because there are multiple variables included in the analysis. Would it be possible to have the y-axis reading be "relative risk of fracture" without the per SD increase? This way the data are more easily interpreted?

*Reviewer #3:*

In this article, Stieglitz et al. present data from the Tsimané of Bolivia demonstrating that in thoracic vertebrae of adults >40 years old, early age at first reproduction and low interbirth interval are associated with lower BMD, and that in Tsimané women but not men, BMD is lower and fracture incidence is higher compared to a control population from the US. The authors argue that this pattern is consistent with tradeoffs in energy allocation between reproduction and somatic maintenance. Overall this is a well written paper and the data will be of significant interest to the field. I have several major comments for the authors to consider.

1) In the Introduction (first paragraph) and Discussion (fourth paragraph), more specificity is needed in the discussions of bone strength. "Bone strength" can refer to 1) midshaft robusticity or cross-sectional geometry (the shape and area of cortical bone in a given plane) vs. 2) bone mineral density (grams of mineral in a given area or volume of bone, which can be measured in cortical or trabecular bone). They are both generally correlated with strength but can't be directly compared to each other. This is important because the background largely focuses on studies of midshaft cortical bone, but the vertebra, which was measured in this study, is primarily trabecular bone surrounded by a thin cortical shell. Note also that the Chirchir references use the word 'density' to refer to trabecular bone volume fraction (BV/TV (%), or the amount of bone in a region of interest) while most publications use density to refer to the amount of bone mineral per area or volume of bone.

2) Introduction, second paragraph, Discussion first and third paragraphs: Osteogenic responses to mechanical loading occur predominantly in the ~2 years before and after puberty, and are markedly attenuated after menarche (although mechanical loading is essential for maintaining existing bone). See review, MacKelvie et al., 2002, http://dx.doi.org/10.1136/bjsm.36.4.250 Also, while cortical bone is highly responsive to mechanical loading around puberty, much less is known about trabecular bone response to mechanical loading. After menarche, physical activity does not provoke a substantial osteogenic response, and consistent with the life history framework presented in this article, energy expended on physical activity might arguably inhibit skeletal recovery (as in the Female Athlete Triad).

3) Previous studies in industrialized populations have looked for relationships between BMD and age at first birth, parity, etc. The authors may wish to cite/discuss Parazzini et al. 1996 J Epidemiol Community Health; We et al. 2018 doi: 10.3346/jkms.2018.33.e311; Chantry et al., 2004; Bexerra et al. 2002 DOI: 10.1093/jn/132.8.2183; etc.

4) The idea that these data reflect tradeoffs in energy allocation to maintenance vs. reproduction is intriguing and makes intuitive sense. It raises a crucial question, however: is energy intake limited in the Tsimané? If not, then why must there be such tradeoffs? Given all the data available to the authors, it would be helpful to identify specific biological mechanisms that are mediating the effects of age at first birth and IBI on thoracic vertebral BMD and fracture incidence in Tsimané vs. American women (or at least frame hypotheses about what they are). For example, what are estradiol levels in Tsimane women, and how do they compare to those of the American women described in this study?

---

## [Author Response]

Summary and essential revisions:Based on the reviewer comments, the discussion of the comments and the editors' assessment, we recommend to revise the manuscript into a version that does not try to test a specific hypothesis, but is more open to different interpretations. The main issue is that the current version favors a life history explanation (which is interesting and compelling), but has not demonstrated that this can be differentiated from other existing explanations. There is already evidence for life history parameters influencing bone mechanics independent of physical activity levels: Evol Med Public Health;2018(1):167-179.

In the revised Introduction and Discussion, we state that the proposed life history interpretation focusing on reproductive effort is not an alternative to and may complement other hypotheses of bone structural variation, including those derived from life history theory (e.g. proposed in Macintosh et al., 2018, which we now cite). We also point out that the proposed life history interpretation

focusing on reproductive effort is unique relative to other explanations (ultimate and proximate) because it focuses on sex differences, both within and between populations, given women’s greater energetic costs of reproduction. Other explanations (e.g. focusing on nutrition, genetics, inflammation, hormones, habitual physical activity level) cannot easily explain the observed population-level sex differences (i.e. Tsimane BMD is lower versus Americans, but only for women, whereas minimal BMD differences exist between Tsimane and American men despite large differences in their physical activity levels). According to these other explanations, population-level differences should be directionally consistent and of similar magnitude for both sexes.

In addition to revising the Introduction and Discussion, we also acknowledge in both sections other proximate explanations which may complement the proposed life history interpretation (i.e. “maternal depletion syndrome”, and factors such as calcium deficiency and/or chronic immune activation which may interact with each other and with reproductive effort).

The full comments of the reviewers are attached to this letter. Please respond to them in detail when submitting the revised version.

Reviewer #1:

[…] The key to evaluating this paper is whether LHT provides a testable alternate hypothesis relative other mechanisms. Does it explain the observed variation or are other mechanisms a parsimonious explanation? Overall this paper presents an important dataset, and I find the analyses appropriate, however the interpretations need further delimitation with regards to other mechanisms.

Thank you, and we agree that further delimitation is needed (see above comments to the editor and more below).

With regards to the analyses of variation within the sample of Tsimane women, the authors interpret the results as being related to life history factors: higher fertility, earlier first reproduction, and shorter interbirth intervals. This is an innovative interpretation of intrapopulation variation in bone mass, but to make the argument that life history explains variation requires an assumption that bone has the capacity to recover from mineral losses associated with pregnancy and lactation. The authors cite a few papers in support of the 'full recovery' of BMD following lactation (Prentice, for example), but I am highly skeptical of this literature. In general, trabecular bone lost during adulthood, for any reason, is never fully regained. The studies used to justify this interpretation are dated and use very coarse methods for the characterization of bone quality. I am not convinced that this literature is sufficient to rule out the Tsimane bone loss as being the cumulative effect of multiple periods of gestation and lactation 'raiding' the skeleton for resources. If there was anything less than 100% recovery with each pregnancy, you would find losses in bone as a simple result of maternal provisioning and the direct 'cause' would not necessarily be interbirth intervals, but simply the cumulative bone loss with subsequent pregnancies due to the skeleton being used as a mechanism of resource storage. This is not necessarily incompatible with life history theory, but it is difficult to see how one would differentiate whether resource depletion during pregnancy would be differentiated from other life history parameters in the current dataset. The more pregnancies a woman has, the shorter the interbirth interval and the less bone might recover. In essence, this itself is a 'life history' trait, of maternal sacrifice of bone quality and reinvestment of these resources in offspring, but it doesn't fit the model and interpretation as the authors present it.

In the comment above reviewer 1 makes three important and related points: 1) that a life history theory (LHT) interpretation requires an assumption, which may not hold, that bone can fully recover from mineral losses associated with pregnancy and lactation; 2) that in the Introduction we cite dated studies using crude methods when referring to maternal skeletal changes associated with pregnancy and lactation to justify a LHT interpretation; and 3) that another mechanism, namely the cumulative effect of multiple periods of gestation and lactation “raiding the skeleton for resources”, can explain Tsimane bone loss without requiring the assumption of full bone mineral recovery following pregnancy and lactation, and that this other mechanism may be consistent with predictions from LHT. In what follows we deal with each of these three points in turn.

Regarding the first point, we agree with reviewer 1 that it is important to clarify whether bone can fully recover from mineral losses associated with pregnancy and lactation. However, we respectfully disagree with the argument that a LHT interpretation requires that bone can fully recover in this manner. LHT posits that energetic resources invested in reproduction should trade-off against resources invested in somatic maintenance under a wide range of conditions. This broad logic – which was not developed to explain variation in bone strength per se – motivates the hypothesis that greater reproductive effort compromises bone strength (where adult bone strength represents an indicator of energetic investment in somatic maintenance). LHT makes no specific predictions about whether bone fully or partially recovers from mineral losses associated with pregnancy and lactation. This is now clarified in the paper (Introduction), and we delete any mention of “full recovery” in this context (there was one such instance in the original manuscript). Reviewer 1 rightly points out that this greater clarity is needed to inform one’s interpretation of our study’s findings, even though this question of whether bone fully or partially recovers from mineral losses associated with pregnancy and lactation is not the empirical focus of our study (most study participants are post-menopausal).

We also appreciate reviewer 1’s skepticism of the relevant bone metabolism literature that we originally cited, and we share this skepticism, in part for reasons reviewer 1 rightly mentions (e.g. loss during adulthood of trabecular structural connectivity is generally irreversible, as lamellar new bone can be added only on existing surfaces; the trabecular network can therefore not be completely restored, even if bone mass increases). This is why we originally cited the Jarjou et al., 2010 study, which found evidence among Gambian women of incomplete restoration of lumbar spine BMD to pre-pregnancy values by 12 months post-partum. In fact, there is a massive literature on maternal skeletal changes during pregnancy, lactation and post-weaning recovery—spanning studies of diverse species (e.g. humans, rats, mice, monkeys), diverse physiological processes (e.g. changes in mineral ions, calciotropic and phosphotropic hormones, sex steroids and other hormones, upregulation of intestinal absorption of calcium and phosphorus, changes in renal mineral handling) and spanning diverse measures of bone quantity and quality (e.g. from DXA, micro-CT, HR-pQCT, microradiography, ultrasound, histology, histomorphometry). A recent comprehensive review of this literature by Kovacs (https://www.ncbi.nlm.nih.gov/pubmed/26887676) includes >1,000 references. The five bullet points below effectively summarize from this review the current understanding of post-weaning skeletal recovery in humans. We apologize if this bullet point summary is verbose, but it directly addresses reviewer 1’s skepticism, and thus merits inclusion in our response. Kovacs writes:

- “The available DXA data suggest that lactational loss of bone density is completely reversed by 12 mo after weaning in most women…whereas recovery has been incomplete at some skeletal sites when assessed at 6 mo or less after weaning…A small longitudinal study of 22 women found a marked loss of 8% in the distal radius that was still reduced 5% by 12 mo after weaning, which suggests that the radius is slower to recover BMD. Note that these studies report the mean responses of a cohort; the response of individual women will vary, such that some women may not return to baseline after lactation, while others may end up with a BMD even higher than prior to pregnancy. Recovery is fast enough that women who breastfeed for 6 mo or more and have a second pregnancy within 18 mo do not have reduced BMD of the spine or hip at the end of the second pregnancy”.

- “HR-pQCT of the radius and ultradistal femur reveal recovery of trabecular microarchitecture and cortical parameters in women who lactate for shorter intervals, but incomplete recovery in women who lactate for longer. However, follow-up in these studies was for less than 6 mo after lactation ended for women who lactated for a longer duration, which is not sufficient time for recovery or to determine if permanent changes in structure resulted. Similar to studies in animals, some clinical studies have found that the cross-sectional diameter of the femur increased after lactation or post-weaning recovery, and that cortical bone area is restored or increased. An increase in bone volumes will improve bone strength and may compensate for any permanent loss of trabecular microarchitecture”.

- “Hip structural analysis of data from DXA scans provides direct measurements of femoral geometry and derived measures of femoral strength. A longitudinal study obtained these measurements at 2 wk, 6 mo, and 12 mo postpartum in lactating women, while similar timing of measurements were carried out in nonpregnant, nonlactating women. In lactating women there was an ~3% decrease in cross-sectional area, a 1.7% decline in cortical thickness, a 2.1% decrease in section modulus (bending strength), and a 2.3% increase in buckling ratio (instability) by 6 mo of lactation. These changes reversed by ~6 mo after lactation ceased, during which no changes were seen in the controls”.

- “If a permanent reduction in BMD or skeletal strength were to occur from lactation, then lactation should be a strong risk factor for low BMD or fracture in women of all ages. However, over five dozen epidemiologic studies of pre-, peri-, and postmenopausal women have found a neutral effect or a protective effect of lactation on peak bone mass, BMD, and fracture risk…There are, of course, a few contrary studies that suggest lactation predicts lower BMD”.

- “Extensive animal data and human data indicate that bone turnover is also increased during the post-weaning phase, but uncoupled in the reverse direction from lactation to favor bone formation. The maternal skeleton undergoes substantial improvements in bone mass, mineralization, and strength. Rodents appear to fully regain mineralization and strength within 2–8 wk, whereas women likely achieve this closer to 1 yr after weaning. In both the animal and human studies, there may be incomplete recovery of the trabecular microarchitecture of the femora and tibiae, the effects of which may be offset by an increase in the cross-sectional diameters of these bones. Although no formal measurements of bone strength after weaning have been done in women, the fact that more than five dozen epidemiological studies found that lactation confers a neutral or protective effect against low BMD and fragility suggests that recovery is effectively complete, without long-term adverse effects on bone strength”.

In the revised Introduction we expand our treatment of this topic and summarize several of these empirical findings (we now refrain from stating that post-weaning restoration of bone strength is necessarily fully complete, or even net positive), and we summarize findings from more recent studies that are not in the Kovacs 2016 review, including Bjornerem et al., 2017 (see Introduction).

Regarding the second point, that we cite dated studies using crude methods when referring to maternal skeletal changes associated with pregnancy and lactation, we have revised the relevant sections of the Introduction (third paragraph) to include more recent citations and studies using higher resolution methods (e.g. HR-pQCT of the appendicular skeleton).

Regarding the third point, that the cumulative effect of multiple periods of gestation and lactation “raiding the skeleton for resources” can explain Tsimane bone loss (without requiring the assumption of full bone mineral recovery following pregnancy and lactation), this logic is consistent with a LHT interpretation and the expected trade-off between energetic investment in reproduction and somatic maintenance. There is no reason to consider this an alternative mechanism that either falls outside the scope of a LHT interpretation or requires a restating of a LHT interpretation. For this reason we discussed in the original Introduction the “maternal depletion syndrome” literature, which focuses precisely on the cumulative long-term effects on maternal health (including bone health) of repeated pregnancies. This literature and its associated logic complements a LHT interpretation, including the shared prediction of an inverse association between lifetime reproductive effort and bone strength (however operationalized), rather than providing an alternative interpretation with different predictions. Prior maternal depletion researchers (e.g. see Figure 1 in Winkvist et al. 1992:692; https://www.ncbi.nlm.nih.gov/pmc/articles/PMC1694126/) have considered different patterns of energy balance (i.e. from positive to negative) over time for a given reproductive cycle, but these researchers have not made specific predictions or assumptions about whether bone tissue in particular fully recovers from mineral losses following specific bouts of pregnancy and lactation. While “maternal depletion syndrome” is characterized by net negative energy balance over time for a given reproductive cycle (line D in Figure 1 in Winkvist et al.), predictions regarding whether bone fully or partially recovers do not directly follow. Interestingly, as we and others (e.g. Kovacs 2016) have noted, many studies show null associations between parity and BMD or fracture risk, and other studies show positive (or negative) associations between parity and BMD. Therefore, in the current literature, evidence of a cumulative negative effect of multiple periods of gestation and lactation on maternal bone strength – ignoring reproductive timing (e.g. age at first birth, IBIs) – is not particularly strong. Evidence of parity-specific effects on bone strength indicators are even mixed within the Tsimane population; in a prior study of Tsimane women (pre- and post-menopausal) we found an inverse association between parity and calcaneal qUS measures (Stieglitz et al. 2015), whereas our present study finds no parity effect on CT-derived thoracic vertebral BMD or fracture risk.

Lastly, reviewer 1 is correct in pointing out the difficulties of differentiating a “pure parity effect” vs. a “pure IBI effect” on bone strength, as higher parity women also have shorter IBIs on average. A strength of our Tsimane dataset is that we can leverage observed variation to explore bone strength indicators for high parity women with shorter vs. longer IBIs; for this reason we included Supplementary figure 3 from the original manuscript, which remains in the main text of the revised manuscript as Figure 1. These results suggest that the association between parity and BMD is complex and merits simultaneous consideration of reproductive timing and relevant physiological mechanisms (the latter fall outside the scope of our paper).

The comparison between the Tsimane and American populations is used to draw more general conclusions about variation in humans, and forms the basis of the most significant inferences made in the paper. Here I find the analyses somewhat problematic. The results of these comparisons demonstrate that Tsimane BMD is lower than the Americans, but only for women; and that Tsimane thoracic vertebral fracture prevalence is higher than Americans. These results are interpreted within the context of life-history theory, and it is argued that the lower BMD and higher facture prevalence among the Tsimane women is associated with greater reproductive effort.

This characterization is accurate. We try to avoid excessive speculation when making general conclusions about human variation based on our study’s results.

The most significant issue is with the characterization of the differences in physical activity levels (PALs) between populations. A few authors have previously discussed broad changes in PAL relative to bone robusticity, but most authors do not use this sort of characterisation. The authors argue that the literature is largely based around the assumption that the transition to agriculture is characterized by a reduction in physical activity levels, and as they consider in the Discussion, this isn't really appropriate. They are correct to point this out in the Discussion, but in setting up the paper around this assumption they are both mischaracterizing most of the literature and setting up a straw man argument. What the literature does suggest is a reduction in terrestrial mobility (lower limb loading) associated with the origins of agriculture in most regions but this is also regionally variable. Change is upper body mechanical loading, however, are much more variable across major dietary transitions, so labour/activity intensity may significantly increase with subsistence transitions in some areas. Virtually no research in the prehistoric biomechanics literature tries to pool upper and lower body (systemic) variation in bone mechanical properties and interpret it in the context of overall PALs. Mischaracterizing the literature this way provides an easy target to knock down. One thing that virtually all biomechanics literature agrees upon is that the mechanical response to loading is variable throughout the body relative to local factors of weight bearing activity and muscle function. Within the context of the current study, it is not clear whether thoracic vertebrae would be expected to track either the mechanics of the lower limb (mobility) or the upper limb (manual labour), or some other more general factors. Many might expect it would track different influences on bone quality such as diet. As the study is pitched, the authors characterize that thoracic BMD should track PALS, and they characterize the literature on terrestrial mobility as being an indicator of PALs. This neither accurately characterizes current knowledge, nor reflects the subtlety of what we do know about bone variation in the past.

We thank reviewer 1 for making two important points. The first point is that a more nuanced perspective is needed when characterizing the prehistoric biomechanics literature, particularly in the Introduction, that considers differential physical activity-induced mechanical loading regimes for lower and upper limbs associated with subsistence transitions. Furthermore, while both lower and upper limb loadings are expected to vary geographically during subsistence transitions, upper limb loading variability is much greater than that for the lower limb; in some areas, upper body activity intensity and loading may actually increase with subsistence transitions, and so we agree that it is misleading and inaccurate to interpret both upper and lower body (i.e. systemic) variation in bone mechanical properties in the context of overall PALs. This more nuanced perspective suggested by reviewer 1 is important because it identifies a flawed assumption – that transition to agriculture caused reductions in PALs – underlying a broader straw man argument.

We regret that in setting up the paper we mischaracterized the prehistoric biomechanics literature on changes in PALs relative to bone strength during subsistence transitions.

Our intention was not to set up a straw man argument, and in fact, a previous version of this manuscript (that wasn’t submitted to *eLife*) included a longer, more detailed description of the relevant prehistoric literature that addressed some of reviewer 1’s comments. Ultimately, because our study does not directly test any hypotheses of human morphological variation in prehistoric populations (noted in the Discussion), we elected to limit mention of the relevant prehistoric literature on subsistence transitions in setting up the empirical focus of our paper. In so doing we gave a mistaken impression, which we now correct. The revised Introduction no longer references changes in PALs relative to bone strength during prehistoric subsistence transitions, nor does it invoke the misguided assumption (or otherwise imply) that prehistoric agricultural transition entailed global PAL reductions and systemic (i.e. lower and upper limb) reductions in bone strength. We also now state in the Introduction that mechanical response of bone to loading is variable throughout the body relative to local factors of weight-bearing activity and muscle function (Introduction, first paragraph). In addition, we now state in the Discussion that during subsistence transitions, changes in upper body activities were likely much more variable than lower body activities, so upper limb mechanical loading may have actually increased with agriculture in some regions (Discussion, last paragraph).

The second point is that it is unclear whether thoracic vertebrae track lower and/or upper limb mechanics, or other factors. Providing this clarity is important for interpreting our study’s results in light of hypotheses of bone structural variation, and for making more general inferences about human bone structural variation. We now clarify in a new paragraph in the Introduction (seventh paragraph) that thoracic vertebrae track both lower and upper limb loadings. We provide information on thoracolumbar loading in the context of activities of daily living, vertebral architecture to accommodate these loads, and changes in architecture with age and sex. Inclusion of this new paragraph provides some justification for our reasoning, which was implied in the original manuscript but never explicitly stated, that thoracic BMD should track both lower and upper limb mechanics.

A third issue is that of the characterization of the Los Angeles comparative group. The variance in biomechanical properties is usually much higher in contemporary populations on account of the internal genetic, dietary and behavioural diversity. This makes characterization of activity extremely difficult. The authors characterize the 'physical inactivity' hypothesis as predicting that modern women from Los Angeles would have lower BMD, but I'm not sure this is necessarily true. Throughout the body of literature on bone biomechanics within recent human populations, declines in bone strength are not always consistent between hunter-gatherers and agricultural populations, and is highly variable throughout the Holocene. Most notably, contemporary human populations have highly variable skeletal strength properties that are dependent upon individual variation in genetics, diet and activity. The range of variation among contemporary populations is typically so high that one rarely finds statistical differences between modern humans and prehistoric populations that are less variable, as the range of variation in the more homogenous populations is found within contemporary society. Imagine, if you will, trying to characterise the 'behaviour' of a typical person in Los Angeles. Could you do it? For any person chosen at random, there is a chance that they may be virtually sedentary, with extremely low PALs, another may be an ultra-marathon runner with extensive athletic training and competition. As a result, it is generally accepted among bone biomechanics researchers that making general assumptions about activity in contemporary populations is challenging at best. This issue could be addressed with any individual activity data for modern subjects, either by recall or activity trackers. Does any such data exist?

Unfortunately we lack individual-level activity data for both the Los Angeles comparative sample and Tsimane study participants (but accelerometry and recall activity data are currently being collected among Tsimane study participants). We now acknowledge this lack of activity data as a study limitation (subsection “Inferring behaviors underlying morphological variation in past human populations”). Reviewer 1 makes several valid points with respect to high documented variability in activity levels and bone strength properties in contemporary populations; due to this high variability, comparisons between contemporary and more homogenous prehistoric populations (e.g. in skeletal strength parameters) often do not reveal statistical differences. We do find statistical differences in BMD across populations, but reviewer 1 rightly points out that it is difficult to characterize activity patterns at the population level. Indeed, Tsimane physical activity levels are not outside the range of some industrialized populations (https://www.ncbi.nlm.nih.gov/pubmed/23383262). Reviewer 1 thus wonders whether our characterization of the physical inactivity hypothesis of skeletal fragility as predicting that women from Los Angeles have lower BMD than Tsimane is accurate.

There is ample evidence that Tsimane are, on average, more active than Los Angeles residents. A telephone survey of 8,354 randomly selected adults in Los Angeles County (http://publichealth.lacounty.gov/ha/reports/habriefs/v3i2_phys/physact.pdf) found that 61% of adults did not get enough weekly physical activity to meet recommended guidelines of the Centers for Disease Control and Prevention, the American College of Sports Medicine (ACSM) and the National Institutes of Health. ACSM guidelines recommend that all adults perform ≥30 minutes of moderate intensity physical activity (defined as the effort expended by an average adult in walking 1.5-2 miles in half an hour) on most and preferably all days – either in a single session or accumulated throughout the day in multiple bouts. Each activity should last at least 8-10 minutes. 41% of LA respondents were classified as “sedentary” and 20% engaged in only irregular activity. Respondents classified as “irregular exercisers” reported vigorous physical activity on 1-3 days and walking for <30 minutes on a typical day or strength training. While the irregular exercise group had some form of physical activity, it was not deemed sufficient to meet current recommendations.

In contrast, based on a large sample of systematic direct behavioral observations (see Gurven et al., 2013), Tsimane activity profiles far exceed ACSM recommendations. Men and women spend about five and two hrs/day, respectively, in food production (i.e. hunting, fishing, foraging and farming). Women and men spend an additional 4–6 hrs and 1.5 hrs/day, respectively, in domestic tasks (e.g. collecting firewood and water, food processing, childcare). By age 15, both sexes are already close to adult work levels in terms of time allocation. About 4–6 hrs/day for women and 6–7 hrs/day for men are spent in lifestyle-moderate activity and relatively little time is spent sedentary. We can also directly compare activity levels – indicated by steps per day from waist-worn accelerometers – between the US NHANES sample (n=1,512 men and 1,545 women) and a different Tsimane sample (i.e. not receiving CT scans, and much larger [n=231 men and 232 women] than the accelerometry sample from the Gurven et al., 2013 paper). There are large population-level differences in the expected direction that are maintained throughout adulthood (see Author response image 1). Unfortunately no comparable data exist yet for children, but we are currently collecting accelerometry data among a large sample of Tsimane children.

For these reasons, even though we cannot make direct activity comparisons between the two samples in our current study, we think it is entirely consistent with available empirical data to reason that Tsimane are, on average (variability notwithstanding), more physically active than Los Angeles residents. We now specify throughout the revised manuscript that our characterization of activity levels across the two populations refers to *mean* differences. We also state that the prediction that Los Angeles residents have lower BMD than Tsimane is a *simple* (i.e. not nuanced) prediction from a physical inactivity hypothesis of skeletal fragility.

The authors note that Tsimane women have moderate to active PAL levels, while men are highly active. Depending on the model of mechanically induced bone remodeling that one accepts, moderate levels of activity may not be sufficient to engender functional adaptive response in the mechanical properties of bone. When compared to the American sample, the specific PAL and reproductive history of the women is unknown. While it is probably safe to assume that the women did have lower fertility and longer inter-birth intervals, it may not be as safe to assume differences in PALs. The Tsimane women have moderate PALs, the American women may have had highly variable PALs with the same population, and activity levels that may have also been 'moderate'. Very little information is provided here about the American data, either in terms of population characteristics and context, or methods, so the reader cannot evaluate whether the characterization of the American group is appropriate.

For Tsimane men and women, PALs (defined as the ratio of total daily energy expenditure to daily basal metabolism) are about 0.1–0.3 units higher than those of industrialized populations (Gurven et al., 2013). A difference of only 0.1 PAL may seem small, but for a typical Tsimane man (62 kg) or woman (56 kg) amounts to an additional 144 and 120 calories expended per day in activity, respectively. As we acknowledge above and in the revised manuscript, a limitation of our study is that we lack activity data (frequency and intensity) at the individual level for both the comparative American and Tsimane samples. However, we can examine previously published activity data from the Multi-Ethnic Study of Atherosclerosis (MESA) sample – which represents one of the American comparative samples in our study (for vertebral fracture prevalence) – to get a sense of American activity patterns. Nearly all MESA participants completed the Typical Week Physical Activity Survey (n=5,829), which was designed to identify time spent in various activities during a “typical week in the past month” (https://www.ncbi.nlm.nih.gov/pubmed/27403323). Only one-third of MESA participants reported any vigorous activity in a typical week, and mean ± SD hours per day in “leisure sedentary behaviour” (i.e. television watching and reading) was 3.4 ± 2.2, including 2.1 ± 1.5 hours/day watching television (see Table 1 in this Joseph et al., 2016 paper). These data are not directly comparable to Tsimane data but nevertheless reveal important population-level differences in activity patterns (Tsimane study participants do not watch [or own] televisions and most do not read recreationally). Instead, Tsimane adults typically spend their leisure time either visiting family or friends, playing soccer or even going fishing or hunting, each of which entails greater activity than watching television or reading. For this reason and those mentioned above, we think our characterization of mean activity differences between populations is appropriate, while acknowledging that we do not characterize sample variances due to lack of individual-level activity data.

In terms of methods, the information on how the American data was collected is insufficient as presented. The only details given are "a matched American sample with directly comparable CT derived indicators of bone strength measured by the same laboratory". Specific details of the equipment and settings are provided for the Tsimane data, were the Los Angeles data collected on the same equipment? With the same settings? Were any objects/participants scanned in each location to validate pooling data from different equipment? This needs to be expanded upon, if only briefly in the text or in supplementary information, so that the reader can evaluate for themselves comparability of approaches.

Thanks for pointing out this oversight. We now provide details in the supplement on how American data were collected (subsection “Thoracic computed tomography (CT)”).

Overall, while my review has been critical of several aspects of the study and interpretations, I would summarize the work more positively. The Tsimane research has provided an extremely important dataset with very interesting social and biological context. The analyses are appropriate and reveal some very interesting trends in bone mass. My concerns with the paper are primarily about interpretation and assumptions relating to activity levels in the American dataset. Both of these concerns mean it is difficult to evaluate the 'life history' hypothesis relative to other proximate mechanisms, or even more broad variation in genetics, habitual activity (and what influence it may have on thoracic vertebrae), early life nutrition, growth plasticity, prevalence of bottle feeding, or extended lactation that may be associated with bmd losses. It is not clear how life history variation necessarily provides a stronger explanation for the data than would any one of these other factors, so alternate explanations should be considered and set up, where possible, as alternate explanations of the data with different predictions of the underlying variation. I believe this dataset and analysis is important and publishable, but it requires more nuanced hypothesis testing to differentiate life history mechanisms from competing hypotheses.

Thank you very much for your thoughtful, wide-ranging and constructive comments. Please refer to the comment entitled “SUMMARY and Essential revisions”) for a more in-depth response to your concerns expressed in the above paragraph.

Reviewer #2:

[…] 1) There are a number of predictions made at the beginning of the manuscript (Introduction) and consequently there are many results to report and discussion to follow. I recommend that the Discussion section follow the order in which the predictions were made so that it is easier for the reader to follow and tie in the predictions and findings.

We appreciate reviewer 2’s helpful recommendation to enhance organization of the Discussion section and paper overall. As recommended, we have restructured the Discussion to more closely follow the order in which the predictions are outlined in the Introduction. To further enhance clarity and organization in the Discussion we added sub-section headings.

2) Additionally, in the Introduction there is extensive discussion about gracilization especially in the fossil/archaeological record, however in the Discussion section the subject is a little glossed over. I would suggest that the implications and/or the complexity of interpreting gracilization from a broader perspective be discussed. I realize that the authors do this e.g., in the Discussion, but a little more could be added as a way to explain/justify the value of this study which goes beyond investigating physical activity levels unlike some of the previous work in the fossil and archaeological record.

Again, we appreciate reviewer 2’s suggestion to improve the Discussion section. Reviewer 2 states that a notable imbalance exists between the Introduction and Discussion in the mention of gracilization in the fossil/archaeological record, with the imbalance favoring the Introduction over the Discussion. In fact, whereas the original Introduction contained ~5 lines of text regarding gracilization in the fossil/archaeological record (half of the first paragraph), the Discussion contains >40 lines of text, which now – after incorporating reviewer 2’s helpful comment #1 above – includes an entire sub-section with three paragraphs. We thus do not feel as though we paid insufficient attention to this issue in the Discussion relative to the Introduction. In the Introduction, when motivating our study’s objectives, we do not pay more attention to gracilization in the fossil/archaeological record because our study’s focus is on contemporary populations, and we cannot directly test hypotheses about behavioral factors underlying gracilization in past human populations. Lastly, we refrained from further discussing gracilization in the Discussion or Introduction because we wanted to respect *eLife*’s suggested maximum word count.

3) While the authors describe how BMD was measured using the computed tomography, I would suggest that a brief description of whether BMD estimates includes both cortical and trabecular bone and if not, how were the two demarcated? The information regarding how the scanner estimates BMD is found in the supporting information section, if there is room, I would include this in the main Materials and methods section. Also, BMD in three thoracic vertebrae was measured, I suppose the BMD reading is the mean of the three and if so, is this a reading provided automated from the scanner or is this done by the researchers manually after the scans are completed?

We thank reviewer 2 for their attention to detail. We now clarify the subsection “Study design and participants”, whether BMD measurements include cortical bone, and we clarify how cortical bone was demarcated from trabecular bone. We also moved additional details on BMD estimation from the supplement to the main Materials and methods section (subsection “Thoracic vertebral bone mineral density (BMD)”), as suggested similarly by reviewer 3. Lastly, we now clarify that BMD was measured manually by the radiologist (subsection “Thoracic vertebral bone mineral density (BMD)”).

4) I found Figure 4 to be a bit confusing to follow and that is because there are multiple variables included in the analysis. Would it be possible to have the y-axis reading be "relative risk of fracture" without the per SD increase? This way the data are more easily interpreted?

This figure shows the bivariate association between BMD and fracture risk; associations are shown for women and men, and are adjusted for potential confounders which are described in the figure caption. Each point estimate shown and its associated 95% CIs are derived from a different regression model whose output is included in the supplement; to enhance clarity, we have modified the figure caption to inform the reader precisely where in the supplement those regression results are located.

Reviewer #3:

[…] 1) In the Introduction (first paragraph) and Discussion (fourth paragraph), more specificity is needed in the discussions of bone strength. "Bone strength" can refer to 1) midshaft robusticity or cross-sectional geometry (the shape and area of cortical bone in a given plane) vs. 2) bone mineral density (grams of mineral in a given area or volume of bone, which can be measured in cortical or trabecular bone). They are both generally correlated with strength but can't be directly compared to each other. This is important because the background largely focuses on studies of midshaft cortical bone, but the vertebra, which was measured in this study, is primarily trabecular bone surrounded by a thin cortical shell. Note also that the Chirchir references use the word 'density' to refer to trabecular bone volume fraction (BV/TV (%), or the amount of bone in a region of interest) while most publications use density to refer to the amount of bone mineral per area or volume of bone.

We appreciate reviewer 3’s concern, and we now specify indicators of bone strength in these manuscript sections. We did not intend to imply that disparate measures of bone strength (e.g. trabecular bone volume fraction and diaphyseal size or shape) can be directly compared to each other, and we fully agree with reviewer 3 that precise terminology is needed to minimize confusion given the numerous ways that researchers measure bone strength. Reviewer 3 is correct in stating that “the background largely focuses on studies of midshaft cortical bone”, whereas our study focuses on vertebrae, which is “primarily trabecular bone surrounded by a thin cortical shell.” The reason for this discrepancy is that, generally speaking, studies of human midshaft cortical bone are more prevalent in the bone functional adaptation literature within anthropology than studies of trabecular bone. While there has been a recent surge in studies of human trabecular bone, including vertebrae (many of which we cite), studies of midshaft cortical bone still dominate this literature (particularly with respect to bone responsivity to mechanical loading, as noted by reviewer 3 in their next comment below). Our study’s background thus reflects our reading of the current state of the relevant literature, rather than reflecting an attempt to motivate our study’s objectives and situate our findings based on invalid direct comparisons with prior studies utilizing disparate methods.

2) Introduction, second paragraph, Discussion first and third paragraphs: Osteogenic responses to mechanical loading occur predominantly in the ~2 years before and after puberty, and are markedly attenuated after menarche (although mechanical loading is essential for maintaining existing bone). See review, MacKelvie et al. 2002, http://dx.doi.org/10.1136/bjsm.36.4.250 Also, while cortical bone is highly responsive to mechanical loading around puberty, much less is known about trabecular bone response to mechanical loading. After menarche, physical activity does not provoke a substantial osteogenic response, and consistent with the life history framework presented in this article, energy expended on physical activity might arguably inhibit skeletal recovery (as in the Female Athlete Triad).

We thank reviewer 3 for providing the MacKelvie et al. review paper on bone responsivity to weight-bearing exercise among children, which we have now read. We fully agree with reviewer 3 that osteogenic responses to mechanical loading are much stronger in childhood vs. adulthood (particularly in the years just before and after puberty), and that among older adults – the focus of our study – the principal somatic effect of mechanical loading is to preserve existing bone rather than add new bone. The marked post-pubertal attenuation of human osteogenic responses to mechanical loading noted by reviewer 3 is evident from observational studies (cross-sectional and longitudinal) and controlled exercise trials. Meta analyses of controlled exercise trials indicate that adult participation in high-impact activities with high-magnitude loading is especially effective at preserving BMD at the lumbar spine and femoral neck (e.g. https://www.ncbi.nlm.nih.gov/pubmed/18981037 and https://www.ncbi.nlm.nih.gov/pubmed/20013013); both of these skeletal sites include substantial amounts of trabecular bone, suggesting that bone responsivity to mechanical loading is not limited to cortical bone. In response to reviewer 3’s comment, we have modified the text in several ways. We eliminated from the entire manuscript the phrase, “osteogenic response to mechanical loading and high PAL”, as it can imply, inaccurately in the present context, a somatic outcome regarding addition of new bone rather than preservation of existing bone. The ensuing revised text (Introduction, third paragraph, Discussion, third paragraph, and subsection “Study limitations”) is consistent with reviewer 3’s comment that under some circumstances “energy expended on physical activity might arguably inhibit skeletal recovery”. In addition, we now describe and cite the Martyn-St James and Carroll meta-analyses of controlled exercise trials mentioned above (Discussion, second paragraph).

3) Previous studies in industrialized populations have looked for relationships between BMD and age at first birth, parity, etc. The authors may wish to cite/discuss Parazzini et al. 1996 J Epidemiol Community Health; We et al. 2018 doi: 10.3346/jkms.2018.33.e311; Chantry et al. 2004; Bexerra et al. 2002 DOI: 10.1093/jn/132.8.2183; etc.

Thank you for providing these references. We have now read each of these papers. In the revised manuscript we now cite the Chantry et al., 2004 paper (Introduction, fourth paragraph) when mentioning potential effects of early-life pregnancy and lactation on later-life bone strength. Our mention of the Chantry paper is part of our more general summary of findings reported in Kovacs’ extraordinary 2016 review (which includes >1,000 references) on changes in maternal bone mineral during pregnancy, lactation and post-weaning.

4) The idea that these data reflect tradeoffs in energy allocation to maintenance vs. reproduction is intriguing and makes intuitive sense. It raises a crucial question, however: is energy intake limited in the Tsimané? If not, then why must there be such tradeoffs? Given all the data available to the authors, it would be helpful to identify specific biological mechanisms that are mediating the effects of age at first birth and IBI on thoracic vertebral BMD and fracture incidence in Tsimané vs. American women (or at least frame hypotheses about what they are). For example, what are estradiol levels in Tsimane women, and how do they compare to those of the American women described in this study?

Our characterization of Tsimane as “energy-limited” refers to their high energy expenditure relative to consumption (Discussion, second paragraph), and the general lack of food storage. High energy expenditure results largely from a combination of their physically active subsistence lifestyle, high infectious burden requiring energetic investments in immune responses (https://www.ncbi.nlm.nih.gov/pubmed/27375044) and high fertility rates. Recently published data from a small sample using the doubly-labeled water technique suggest a median total energy expenditure for Tsimane women and men of 2,186 and 3,065 kcal/day, respectively (n=20/sex) (https://www.ncbi.nlm.nih.gov/pubmed/27375044). Per-capita energy intake for Tsimane women and men is 2,422 and 2,736 kcal/day (https://www.ncbi.nlm.nih.gov/pubmed/30383188). These estimates were collected in different Tsimane sub-samples and can vary by season and other factors (e.g. pregnancy or lactation status), but nonetheless suggest that Tsimane (particularly men) are often in negative energy balance or only minimally achieve energetic surplus. Tsimane food insecurity is also relatively common (https://www.ncbi.nlm.nih.gov/books/NBK242449/), which reflects in part residential detachment from the market economy. For these reasons we characterize Tsimane as energy-limited compared to industrialized populations.

With respect to identification of specific physiological mechanisms mediating effects of reproductive effort on thoracic vertebral BMD and fracture, there are a plethora of candidates reviewed by Kovacs 2016. These include mineral ions and calciotropic and phosphotropic hormones, parathyroid hormone and parathyroid hormone-related protein, calcitriol and calcidiol, calcitonin, fibroblast growth factor-23, sex steroids and other hormones, upregulation of intestinal absorption of calcium and phosphorus, altered renal mineral handling, and other biochemical markers of bone turnover. At present we either have none of these relevant data for Tsimane, or we cannot directly compare Tsimane vs. American levels (e.g. because assays were performed in different labs using different protocols). In his comprehensive 2016 review, Kovacs notes that relevant physiological mechanisms such as estradiol (since it was mentioned by reviewer 3) operate in an interactive fashion (e.g. with parathyroid hormone-related protein), thus making it difficult to isolate the independent effect of each mechanism. Kovacs also writes, “It is difficult to analyze the independent effects of low estradiol on bone turnover during lactation. More intense lactation causes more breast milk output and bone loss, but more intense lactation is also associated with lower estradiol and prolonged amenorrhea. The impact of isolated low estradiol is that it causes 1–2% annual losses of BMD in recently menopausal women, but the older ages of these women may not reflect what happens to reproductive age women with low estradiol.” These physiological processes are obviously complex and in our view outside the scope of the present study. We thus prefer to avoid speculation regarding how physiological mediators may operate between populations.